METHODS

# Topological state-space estimation of functional human brain networks

**Moo K. Chung**[1]*, **Shih-Gu Huang**[2], **Ian C. Carroll**[3], **Vince D. Calhoun**[4], **H. Hill Goldsmith**[5]

**1** Department of Biostatistics and Medical Informatics, University of Wisconsin, Madison, Wisconsin, United States of America, **2** PPG/ECG Signal, Taipei, Taiwan, **3** Department of Child and Adolescent Psychiatry, New York University Grossman School of Medicine, New York, United States of America, **4** Tri-Institutional Center for Translational Research in Neuroimaging and Data Science (TReNDS), Georgia State University, Georgia Institute of Technology, Emory University, Atlanta, Georgia, United States of America, **5** Department of Psychology & Waisman Center, University of Wisconsin, Madison, Wisconsin, United States of America

* mkchung@wisc.edu

## Abstract

We introduce an innovative, data-driven topological data analysis (TDA) technique for estimating the state spaces of dynamically changing functional human brain networks at rest. Our method utilizes the Wasserstein distance to measure topological differences, enabling the clustering of brain networks into distinct topological states. This technique outperforms the commonly used k-means clustering in identifying brain network state spaces by effectively incorporating the temporal dynamics of the data without the need for explicit model specification. We further investigate the genetic underpinnings of these topological features using a twin study design, examining the heritability of such state changes. Our findings suggest that the topology of brain networks, particularly in their dynamic state changes, may hold significant hidden genetic information.

**Data Availability Statement:** The data used in the study is publicly available through National Data Archive (https://nda.nih.gov) under the collection tile Validating RDoC for Children and Adolescents: A Twin Study with Neuroimaging #2105. The data

## Author summary

The paper introduces a new data-driven topological data analysis (TDA) method for studying dynamically changing human functional brain networks obtained from the resting-state functional magnetic resonance imaging (rs-fMRI). Leveraging persistent homology, a multiscale topological approach, we present a framework that incorporates the temporal dimension of brain network data. This allows for a more robust estimation of the topological features of dynamic brain networks.

The method employs the Wasserstein distance to measure the topological differences between networks and demonstrates greater efficiency and performance than the commonly used *k*-means clustering in defining the state spaces of dynamic brain networks. Our method maintains robust performance across different scales and is especially suited for dynamic brain networks.

In addition to the methodological advancement, the paper applies the proposed technique to analyze the heritability of overall brain network topology using a twin study design. The study investigates whether the dynamic pattern of brain networks is a

can be also obtained by contacting Wisconsin Twin Research (https://goldsmithtwins.waisman.wisc.edu) through email wisconsintwins@waisman.wisc.edu. MATLAB code for the method is available at https://github.com/laplcebeltrami/PH-STAT.

**Funding:** This study was supported by the National Institutes of Health (EB022856, MH133614 to MC; MH101504, P30HD003352, U54HD09025 to HG) and the National Science Foundation (MDS-2010778 to MKC; 2112455 to VC). The funders had no role in study design, data collection and analysis, decision to publish, or preparation of the manuscript.

**Competing interests:** The authors have declared that no competing interests exist.

genetically influenced trait, an area previously underexplored. By examining the state change patterns in twin brain networks, we make significant strides in understanding the genetic factors underlying dynamic brain network features. Furthermore, the paper makes its method accessible by providing MATLAB codes, contributing to reproducibility and broader application.

# 1 Introduction

In standard graph theory-based network analysis, network features such as node degrees and clustering coefficients are obtained from adjacency matrices after thresholding weighted edges [1–4]. The final statistical analysis results can vary depending on the choice of threshold or parameter [5, 6]. This variability underscores the need for a multiscale network analysis framework that provides consistent results and interpretation, regardless of the choice of parameter. Persistent homology, a branch of algebraic topology, presents a novel solution to this challenge of multiscale analysis [7]. Unlike traditional graph theory approaches that analyze networks at a single fixed scale, persistent homology examines networks across multiple scales. It identifies topological features that remain persistent and are robust against different scales and noise perturbations [8–11].

Recent studies have illustrated the versatility of persistent homology in analyzing complex networks, including brain networks. [10, 12] highlighted the application of persistent homology in evaluating temporal changes in topological network features. [13] used persistent homology to detect and track the evolution of networks' clique. [14] discussed the use of simplicial complexes encoded by persistent homology for brain networks. [9] applied persistent homology to investigate the spatial distributions of cliques and cycles in brain networks. [15, 16] showed how persistent homology could be used in the analysis of functional brain connectivity using EEG. [17] utilized persistent homology to analyze brain networks for studying abnormal white matter in maltreated children. These studies collectively emphasize the potential of persistent homology in providing a robust framework for multiscale network analysis. This approach's ability to capture topological features across different scales and under varying conditions makes it particularly suitable for studying the complex brain networks.

Persistent homological network approaches have shown to be more robust and outperform many existing graph theory measures and methods. In [6, 18], persistent homology was shown to outperform eight existing graph theory features, such as clustering coefficient, small-worldness, and modularity. [19] showed persistent homology-based measures can provide more significant group difference and better classification performance compared to standard graph-based measures that characterize small-world organization and modular structure. In [20, 21], persistent homology was shown to outperform various matrix norm-based network distances. In [22], persistent homology was shown to outperform the power spectral density and local variance methods. In [23], persistent homology was shown to outperform topographic power maps. In [24], center persistency was shown to outperform the network-based statistic and element-wise multiple corrections. In [17], persistent homology based clustering is shown to outperform $k$-means clustering and hierarchical clustering. However, the method has been mainly used on *static* networks or a static summary of time-varying networks. The dynamic pattern of persistent homology for time-varying brain networks was rarely investigated, with a few exceptions [9, 17, 25–28].

While Euclidean loss remains the dominant cost function in deep learning, topological losses based on persistent homology are emerging as superior in tasks requiring topological

understanding [29–32]. These topological losses incorporate penalties based on the topological features of the data, distinguishing them from the Euclidean loss, which primarily focuses on differences at the node or edge level. By encoding the intrinsic topological structure of the network, topological losses facilitate the creation of more informative feature maps, potentially enhancing overall model performance [33]. [31] demonstrated that image segmentation based on topological loss outperforms other deep learning architectures for similar tasks. [32] introduced a new architecture that excels in segmenting curvilinear structures by learning topological similarities over existing methods.

In this paper, we propose to develop a novel *dynamic persistent homology* framework for time varying network data. Our coherent scalable framework for the computation is based on the Wasserstein distance between persistent diagrams, which provides the topological profile of data into 2D scatter plots. We directly establish the relationship between the Wasserstein distance and edge weights in networks making the method far more accessible and adaptable. We achieve $\mathcal{O}(n \log n)$ run time in most graph manipulation tasks such as matching and averaging. Such scalable computation enables us to perform a computationally demanding task such as topological clustering with ease. The method is applied in the determination of the state space of dynamically changing functional brain networks obtained from the resting-state functional magnetic resonance imaging (rs-fMRI). We will show that the proposed method based on the Wasserstein distance can capture the topological patterns that are consistently observed across different time points.

The Wasserstein distance or Kantorovich–Rubinstein metric, as originally defined between probability distributions, can be used to measure topological differences [34–36]. Due to the connection to the optimal mass transport, which enjoys various optimal properties, the Wasserstein distance has been applied to various imaging applications. Nonetheless, its application in network data analysis remains relatively limited [17, 37]. [38] used the Wasserstein distance in resampling brain surface meshes. [39, 40] used the Wasserstein distance in classifying brain cortical surface shapes. [41] used the Wasserstein distance in building generative adversarial networks. [42] used the Wasserstein distance for manifold regression problems in the space of positive definite matrices for the source localization problem in EEG. [43] used the Wasserstein distance in predicting Alzheimer's disease progression in magnetoencephalography (MEG) brain networks. [44] enhanced images by regularizing with the Wasserstein distance. However, the Wasserstein distance in these applications is all geometric in nature.

We applied the method to dynamically changing twin brain networks obtained from the resting-state functional magnetic resonance imaging (rs-fMRI). We investigated if the state change pattern in time varying brain networks is genetically heritable for the first time. This is not yet reported in existing literature. Monozygotic (MZ) twins share 100% of genes while dizygotic (DZ) twins share 50% of genes [45]. MZ-twins are more similar or concordant than DZ-twins for cognitive aging and dysfunction [46]. The difference between MZ- and DZ-twins directly quantifies the extent to which imaging phenotypes, behaviors and cognitions are influenced by genetic factors [47]. If MZ-twins show more similarity on a given trait compared to DZ-twins, this provides a piece of evidence that genes significantly influence that trait. Even twin studies on normal subjects are useful for understanding the extent to which psychological and medical disorders, as well as behaviors and traits, are influenced by genetic factors. This information can be used to develop better ways to prevent and treat disorders and maladaptive behaviors. Some of the most effective treatments for medical disorders have been identified as a result of twin studies [48].

Even though there are numerous twin imaging studies, almost all previous studies used *static* univariate imaging phenotypes such as cortical thickness [49], fractional anisotropy [50], functional activation [51–53] in determining heritability in brain networks. There have been a

limited number of studies investigating the heritability of the *dynamics* of brain networks [51, 54]. *It is not even clear the dynamic pattern itself is a heritable trait.* We propose to tackle this challenge. Measures of network dynamics are worth investigating as potential phenotypes that indicate the genetic risk for neuropsychiatric disorders [55]. Determining the extent of heritability of dynamic pattern is the first necessary prerequisite for identifying dynamic network phenotypes.

One of the earliest papers on functional brain activation in twins is based on the resting-state EEG [56], where they observed high twin correlation in MZ-twins on EEG spectra. [52] reported a heritability of 0.42 for default-mode network (DMN) in an extended pedigree study without twins. [57] reported a heritability of 0.54 for DMN on using 24 pairs of MZ and 22 pairs of DZ. [58] studied 79 MZ twins and 46 DZ twin pairs, reporting heritability in only one specific connection: They found statistically significant heritability of 0.41 for the connection between the precuneus and the right inferior parietal/temporal cortex, using a structural equation model. We report far stronger results with much higher heritability in a larger twin study.

## 2 Methods

### Ethics statement

The ethics approval for using the data was obtained from the local Institutional Review Boards (IRB) of University of Wisconsin-Madison (https://irb.wisc.edu). Informed written consent was obtained from all participants.

### 2.1 Graphs as simplicial complices

The proposed method for estimating topological state space is based on the topological clustering on a collection of graphs (Fig 1). The initial step involves a birth-death decomposition of a weighted graph, leading to the generation of sorted birth and death sets (section 2.3). The second step entails calculating the topological distance: between birth sets to obtain the 0D topological distance $d_0$, and between death sets to obtain the 1D topological distance $d_1$ (section 2.4). The third step involves computing the within-cluster distance $l_W$ among the collection of graphs (section 2.6). Subsequently, we demonstrate the equivalence of topological clustering with $k$-means clustering in a high-dimensional convex set, employing $k$-means clustering routines for optimization. To increase the reproducibility, MATLAB codes for performing the methods are provided in https://github.com/laplcebeltrami/PH-STAT.

A high dimensional object such as a brain network can be modeled as weighted graph $\mathcal{X} = (V, w)$ consisting of node set $V$ indexed as $V = \{1, 2, \cdots, p\}$ and edge weights $w = (w_{ij})$ between nodes $i$ and $j$. If we order the edge weights in the increasing order, we have the sorted edge weights:

$$\min_{j,k} w_{jk} = w_{(1)} < w_{(2)} < \cdots < w_{(q)} = \max_{j,k} w_{jk}, \tag{1}$$

where $q \leq (p^2 - p)/2$. The subscript $_{()}$ denotes the order statistic. In terms of sorted edge weight set $W = \{w_{(1)}, \cdots, w_{(q)}\}$, we may also write the graph as $\mathcal{X} = (V, W)$. If we connect nodes following some criterion on the edge weights, they will form a simplicial complex which will follow the topological structure of the underlying weighted graph [7, 59]. Note that the $k$-simplex is the convex hull of $k + 1$ points in $V$. A simplicial complex is a finite collection of simplices such as points (0-simplices), lines (1-simplices), triangles (2-simplices) and higher dimensional counter parts.

The *Rips complex* $\mathcal{X}_\epsilon$ is a simplicial complex, whose $k$-simplices are formed by $(k + 1)$ nodes which are pairwise within distance $\epsilon$ [60]. While a graph has at most 1-simplices, the Rips

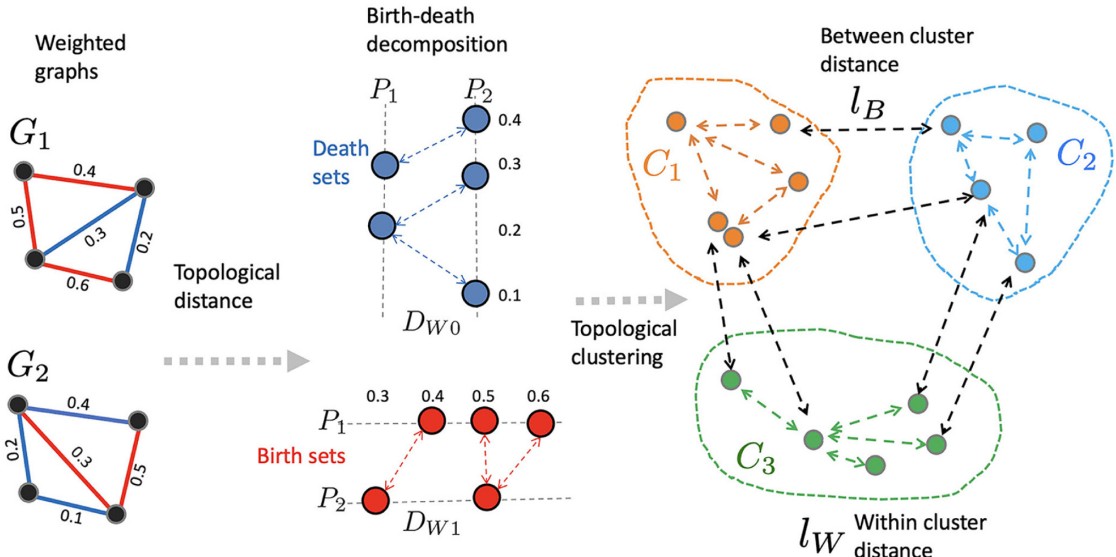

**Fig 1. Proposed topological clustering pipeline used in estimating the state space.** Given two weighted graphs $G_1$, $G_2$, we first perform the birth-death decomposition and partition the edges into sorted birth and death sets (section 2.3). The 0D topological distance $D_{W0}$ between birth values quantifies discrepancies in connected components (section 2.4). The 1D topological distance $D_{W1}$ between death values quantifies discrepancies in cycles (section 2.4). The combined distance $\mathcal{D} = D_{W0}^2 + D_{W1}^2$ is used in computing the within-cluster distance $l_W$ between graphs. Topological clustering is performed by minimizing $l_W$ over all possible cluster labels $C_1, \cdots, C_k$ (section 2.6).

complex has at most $(p-1)$-simplices. The Rips complex induces a hierarchical nesting structure called the Rips filtration

$$\mathcal{X}_{\epsilon_0} \subset \mathcal{X}_{\epsilon_1} \subset \mathcal{X}_{\epsilon_2} \subset \cdots$$

for $0 = \epsilon_0 < \epsilon_1 < \epsilon_2 < \cdots$, where the sequence of $\epsilon$-values are called the filtration values. The filtration is quantified through a topological basis called *k-cycles*. 0-cycles are the connected components, 1-cycles are 1D closed paths or loops while 2-cycles are 3-simplices (tetrahedron) without interior. Any $k$-cycle can be represented as a linear combination of basis $k$-cycles. The Betti number $\beta_k$ counts the number of independent $k$-cycles. During the Rips filtration, the $i$-th $k$-cycle is born at filtration value $b_i$ and dies at $d_i$. The collection of all the paired filtration values

$$P(\mathcal{X}) = \{(b_1, d_1), \cdots, (b_q, d_q)\}$$

displayed as 1D intervals is called the *barcode* and displayed as a 2D scatter plot is called the *persistent diagram*. Since $b_i < d_i$, the scatter plot in the persistent diagram are displayed above the line $y = x$ line by taking births in the $x$-axis and deaths in the $y$-axis.

For a dynamically changing brain network $\mathcal{X}(t) = (V, w(t))$, we assume the node set is fixed while edge weights are changing over time $t$. If we build persistent homology at each fixed time, the resulting barcode is also time dependent:

$$P(\mathcal{X}(t)) = \{(b_1(t), d_1(t)), \cdots, (b_q(t), d_q(t))\}.$$

## 2.2 Graph filtrations

As the number of nodes $p$ increases, the resulting Rips complex becomes increasingly dense. Additionally, as the filtration values rise, the number of edges connecting each pair of nodes also increases, leading to a more interconnected structure. At higher filtration values, Rips filtration becomes an ineffective representation of networks. To remedy this problem, graph filtration was introduced [6, 18]. Given weighted graph $\mathcal{X} = (V, w)$ with edge weight $w = (w_{ij})$, the binary network $\mathcal{X}_\epsilon = (V, w_\epsilon)$ is a graph consisting of the node set $V$ and the binary edge weights $w_\epsilon = (w_{\epsilon,ij})$ given by

$$w_{\epsilon,ij} = \begin{cases} 1 & \text{if } w_{ij} > \epsilon; \\ 0 & \text{otherwise.} \end{cases}$$

Note $w_\epsilon$ is the adjacency matrix of $\mathcal{X}_\epsilon$, which is a simplicial complex consisting of 0-simplices (nodes) and 1-simplices (edges) [60]. While the binary network $\mathcal{X}_\epsilon$ has at most 1-simplices, the Rips complex can have at most $(p-1)$-simplices. By choosing threshold values at sorted edge weights $w_{(1)}, w_{(2)}, \cdots, w_{(q)}$, we obtain the sequence of nested graphs [5]:

$$\mathcal{X}_{w_{(1)}} \supset \mathcal{X}_{w_{(2)}} \supset \cdots \supset \mathcal{X}_{w_{(q)}}. \tag{2}$$

The sequence of such a nested multiscale graph is called the *graph filtration* [6, 18]. Note that $\mathcal{X}_{w_{(1)}-\epsilon}$ is the complete weighted graph for any $\epsilon > 0$. On the other hand, $\mathcal{X}_{w_{(q)}}$ is the node set $V$. By increasing the threshold value, we are thresholding at higher connectivity; thus more edges are removed.

For dynamically changing brain networks (Fig 2), we can similarly build time varying graph filtrations at each time point $\{\mathcal{X}_\epsilon(t) : t \in \mathbb{R}^+\}$.

## 2.3 Birth-death decomposition

Unlike the Rips complex, there are no higher dimensional topological features beyond the 0D and 1D topology in graph filtration. The 0D and 1D persistent diagrams $(b_i, d_i)$ tabulate the life-time of 0-cycles (connected components) and 1-cycles (loops) that are born at the filtration value $b_i$ and die at value $d_i$, respectively. The 0th Betti number $\beta_0(w_{(i)})$ counts the number of 0-cycles at filtration value $w_{(i)}$ and can be shown to be non-decreasing over filtration (Fig 3) [21]:

$$\beta_0(w_{(i)}) \leq \beta_0(w_{(i+1)}).$$

On the other hand the 1st Betti number $\beta_1(w_{(i)})$ counts the number of independent loops and can be shown to be non-increasing over filtration [21]:

$$\beta_1(w_{(i)}) \geq \beta_1(w_{(i+1)}).$$

During the graph filtration, when new components is born, they never die. Thus, 0D persistent diagrams are completely characterized by birth values $b_i$ only. Loops are viewed as already born at $-\infty$. Thus, 1D persistent diagrams are completely characterized by death values $d_i$ only. We can show that the edge weight set $W$ can be partitioned into 0D birth values and 1D death values [61]:

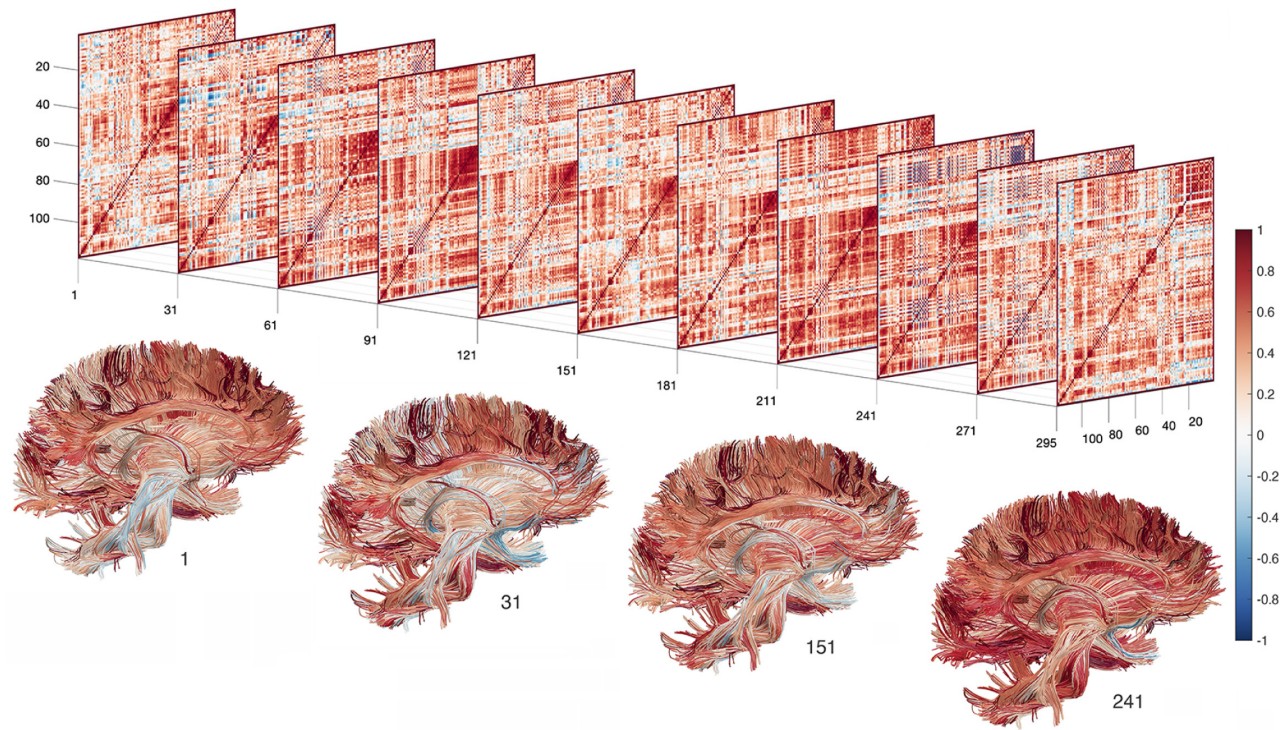

**Fig 2. Dynamically changing correlation matrices computed from rs-fMRI using the sliding window of size 60 for a subject.** The constructed correlation matrices are superimposed on top of the white matter fibers in the MNI space and color coded based on correlation values.

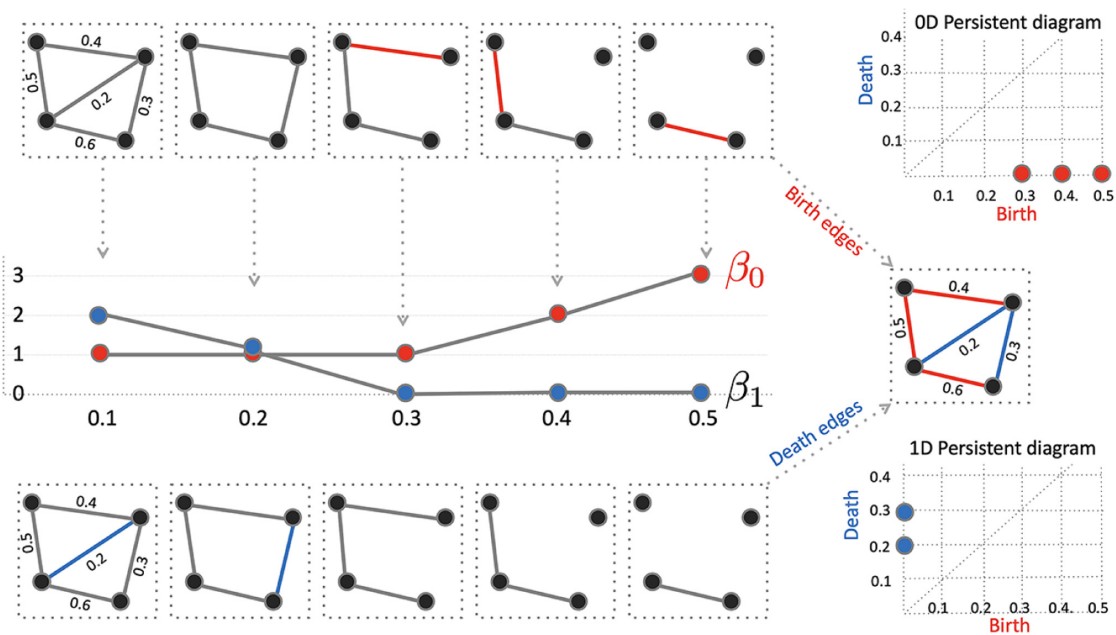

**Fig 3. The birth-death decomposition partitions the edge set into the birth and death edge sets.** The birth set forms the maximum spanning tree (MST) and contains edges that create connected components (0D topology). The death set contains edges that do not belong to the maximum spanning tree (MST) and destroys loops (1D topology).

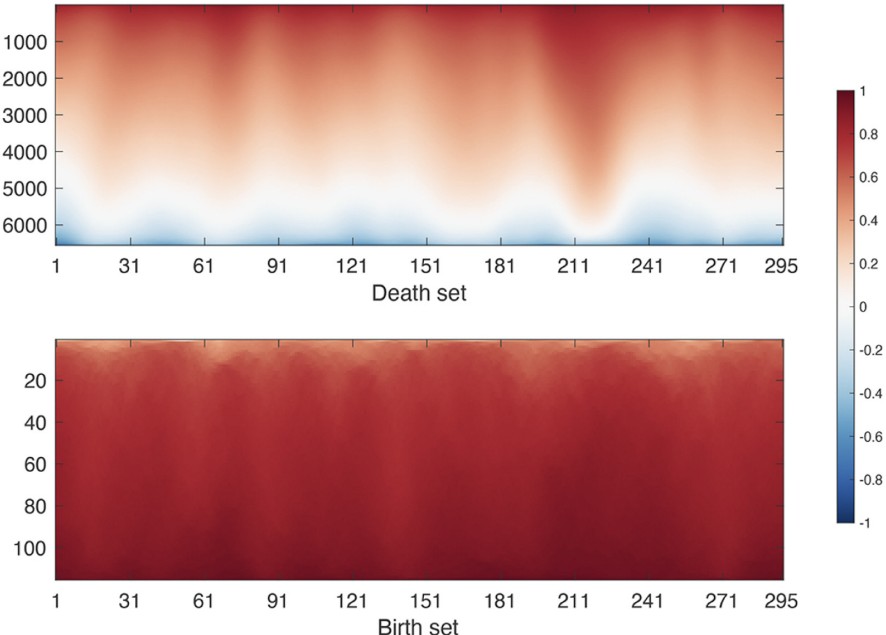

**Fig 4. The corresponding birth and death sets of dynamically changing correlation matrix shown in Fig 2.** The horizontal axis is the time point. Columns are the sorted birth and death edge values at the time point.

**Theorem 1 (Birth-death decomposition).** *The edge weight set $W = \{w_{(1)}, \cdots, w_{(q)}\}$ has the unique decomposition*

$$W = W_b \cup W_d, \quad W_b \cap W_d = \emptyset, \tag{3}$$

*where birth set $W_b = \{b_{(1)}, b_{(2)}, \cdots, b_{(q_0)}\}$ is the collection of 0D sorted birth values and death set $W_d = \{d_{(1)}, d_{(2)}, \cdots, d_{(q_1)}\}$ is the collection of 1D sorted death values with $q_0 = p - 1$ and $q_1 = (p - 1)(p - 2)/2$. Further $W_b$ forms the 0D persistent diagram while $W_d$ forms the 1D persistent diagram.*

In a complete graph with $p$ nodes, there are $q = p(p - 1)/2$ unique edge weights. There are $q_0 = p - 1$ number of edges that produce 0-cycles. This is equivalent to the number of edges in the maximum spanning tree (MST) of the graph. Thus,

$$q_1 = q - q_0 = \frac{(p - 1)(p - 2)}{2}$$

number of edges destroy loops. The 0D persistent diagram is given by $\{(b_{(1)}, \infty), \cdots, (b_{(q_0)}, \infty)\}$. Ignoring $\infty$, $W_b$ is the 0D persistent diagram. The 1D persistent diagram is given by $\{(-\infty, d_{(1)}), \cdots, (-\infty, d_{(q_1)})\}$. Ignoring $-\infty$, $W_d$ is the 1D persistent digram. We can show that the birth set is the MST (Fig 3) [62].

The identification of $W_b$ is based on the modification to Kruskal's or Prim's algorithm that identifies the MST [6, 62]. Then $W_d$ is identified as $W \setminus W_d = W \cap W_d^c$. Fig 4 displays how the birth and death sets change over time for a single subject used in the study. Given edge weight matrix $W$ as input, the Matlab function `WS_decompose.m` outputs the birth set $W_b$ and the death set $W_d$.

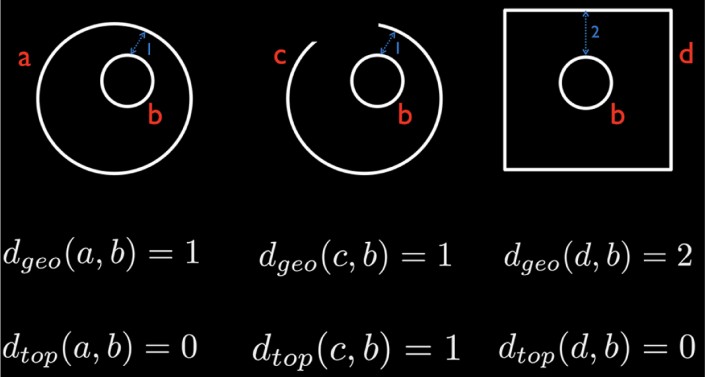

**Fig 5. Comparison between geometric distance $d_{geo}$ and topological distance $d_{top}$.** We used the shortest Euclidean distance between objects as the geometric distance. The left (two circles) and middle (circle and arc) objects are topologically different while the left and right (square and circle) objects are topologically equivalent. The geometric distance cannot discriminate topologically different objects (left and middle) and produces false negatives. The geometric distance incorrectly discriminates topologically equivalent objects (left and right) and produces false positive.

## 2.4 Topological distances

Like the majority of clustering methods such as *k*-means and hierarchical clustering that use geometric distances [6, 63, 64], we propose to develop a topological clustering method using topological distances (Fig 5). For this purpose we use the Wasserstein distance.

Given two probability distributions $X \sim f_1$ and $Y \sim f_2$, the *r-Wasserstein distance $D_W$*, the probabilistic version of the optimal transport, is defined as

$$D_W(f_1, f_2) = \left( \inf \mathbb{E}|X - Y|^r \right)^{1/r}, \tag{4}$$

where the infimum is taken over every possible joint distribution of *X* and *Y*. The Wasserstein distance is the optimal expected cost of transporting points generated from $f_1$ to those generated from $f_2$ [35]. There are numerous distances and similarity measures defined between probability distributions such as the Kullback-Leibler (KL) divergence and the mutual information [65]. While the Wasserstein distance is a metric satisfying positive definiteness, symmetry, and triangle inequality, the KL-divergence and the mutual information are not metrics. Although they are easy to compute, the biggest limitation of the KL-divergence and the mutual information is that the two probability distributions must be defined on the same sample space. If the two distributions do not have the same support, it may be difficult to even define the distance between them. If $f_1$ is discrete while $f_2$ is continuous, it is difficult to define them. On the other hand, the Wasserstein distance can be computed for any arbitrary distributions that may not have the common sample space, making it extremely versatile.

Consider persistent diagrams $P_1$ and $P_2$ given by

$$P_1 : x_1 = (b_1^1, d_1^1), \cdots, x_q = (b_q^1, d_q^1), \quad P_2 : y_1 = (b_1^2, d_1^2), \cdots, y_q = (b_q^2, d_q^2).$$

Their empirical distributions are given in terms of Dirac-Delta functions

$$f_1(x) = \frac{1}{q} \sum_{i=1}^{q} \delta(x - x_i), \quad f_2(y) = \frac{1}{q} \sum_{i=1}^{q} \delta(y - y_i).$$

Then we can show that the *r-Wasserstein distance* on persistent diagrams is given by

$$D_W(P_1, P_2) = \inf_{\psi: P_1 \to P_2} \left( \sum_{x \in P_1} \| x - \psi(x) \|^r \right)^{1/r} \tag{5}$$

over every possible bijection $\psi$, which is a permutation, between $P_1$ and $P_2$ [34–36]. Optimization (5) is the standard assignment problem, which is usually solved by the Hungarian algorithm in $\mathcal{O}(q^3)$ [66]. However, for graph filtration, the distance can be computed *exactly* in $\mathcal{O}(q \log q)$ by simply matching the order statistics on the birth or death values [61, 62, 67]:

**Theorem 2**. *The r-Wasserstein distance between the 0D persistent diagrams for graph filtration is given by*

$$D_{W0}(P_1, P_2) = \left[ \sum_{i=1}^{q_0} (b^1_{(i)} - b^2_{(i)})^r \right]^{1/r}, \tag{6}$$

*where $b^j_{(i)}$ is the i-th smallest birth values in persistent diagram $P_j$. The r-Wasserstein distance between the 1D persistent diagrams for graph filtration is given by*

$$D_{W1}(P_1, P_2) = \left[ \sum_{i=1}^{q_1} (d^1_{(i)} - d^2_{(i)})^r \right]^{1/r}, \tag{7}$$

*where $d^j_{(i)}$ is the i-th smallest death values in persistent diagram $P_j$.*

The proof is provided in [68]. We can show that the 2-Wasserstein distance is equivalent to the Euclidean distance within a certain convex set. Let $\mathbf{b}_i = (b^i_{(1)}, b^i_{(2)} \ldots, d^i_{(q_0)})^\top$ be the vector of sorted birth values of persistent diagram $P_i$. Then $\mathbf{b}_i$ is a point in the $(q_0 - 1)$-simplex $\mathcal{T}_0$ given by

$$\mathcal{T}_0 = \{(x_1, x_2, \cdots, x_{q_0}) | x_1 < x_2 < \cdots < x_{q_0}\} \subset \mathbb{R}^{q_0},$$

where $x_1$ and $x_{q_0}$ are bounded below and above respectively. If brith and death values are from correlation matrices, $-1 \leq x_1$ and $x_{q_0} \leq 1$. Hence, the 0D Wasserstein distance is equivalent to Euclidean distance in the $q_0$-dimensional convex set $\mathcal{T}_0$. Similarly, the vector of sorted death values $\mathbf{d}_i = (d^i_{(1)}, d^i_{(2)} \cdots, d^i_{(q_1)})^\top$ of persistent diagram $P_i$ is a point in the $(q_1 - 1)$-simplex $\mathcal{T}_1$ given by

$$\mathcal{T}_1 = \{(x_1, x_2, \cdots, x_{q_1}) | x_1 < x_2 < \cdots < x_{q_1}\} \subset \mathbb{R}^{q_1}, \tag{8}$$

where $x_1$ and $x_{q_1}$ are bounded below and above respectively. Hence, the 1D Wasserstein distance is equivalent to Euclidean distance in the $q_1$-dimensional convex set $\mathcal{T}_1$.

## 2.5 Topological mean and variance

Given a collection of graphs $\mathcal{X}_1 = (V, w^1), \cdots, \mathcal{X}_n = (V, w^n)$ with edge weights $w^k = (w^k_{ij})$, the usual approach for obtaining the average network $\bar{\mathcal{X}}$ is simply averaging the edge weight matrices in an element-wise fashion

$$\bar{\mathcal{X}} = \left( V, \frac{1}{n} \sum_{k=1}^{n} w^k_{ij} \right).$$

However, such average is the average of the connectivity strength. Such an approach is usually sensitive to topological outliers [21]. We address the problem through the Wasserstein distance. A similar concept was proposed in the persistent homology literature through the Wasserstein barycenter [69, 70], which is motivated by the Fréchet mean [71–74]. However, the method has not seen many applications in modeling graphs and networks.

To account for both 0D and 1D topological differences in networks, we use the sum of 0D and 1D Wasserstein distances between networks $\mathcal{X}_1$ and $\mathcal{X}_2$ as the topological distance

$$\mathcal{D}(\mathcal{X}_1, \mathcal{X}_2) = D_{W0}^2(P_1, P_2) + D_{W1}^2(P_1, P_2). \tag{9}$$

The equal weights of the form (9) were used for the following reasons. Through the birth-death decomposition, a weighted graph can be topologically characterized by 0D and 1D features, with no higher-dimensional features present. However, it is unclear which feature contributes the most. Equal weighting of 0D and 1D features ensures a balanced representation without bias towards either type of feature.

Let $\varnothing$ denote a graph with zero edge weights. Then, due to the birth-death decomposition, we have

$$\mathcal{D}(\mathcal{X}_1, \varnothing) = \sum_{i<j} (w_{ij}^1)^2, \quad \mathcal{D}(\varnothing, \mathcal{X}_2) = \sum_{i<j} (w_{ij}^2)^2,$$

where the squared sums of all the edge weights make the interpretation straightforward. If unequal weighting is used, these relationships do not hold. Further, the Wasserstein distances $D_{W0}$ and $D_{W1}$ are equivalent to the Euclidean distances in a convex set. Therefore, squared distance is a more natural choice that satisfies the triangle inequality

$$\mathcal{D}(\mathcal{X}_1, \mathcal{X}_3) \leq \mathcal{D}(\mathcal{X}_1, \mathcal{X}_2) + \mathcal{D}(\mathcal{X}_2, \mathcal{X}_3),$$

thus qualifying as a metric.

The sum (9) does not uniquely define networks. Like the toy example in Fig 5, we can have many topologically equivalent brain networks that give the identical distance. Thus, the average of two graphs is also not uniquely defined. The situation is analogous to Fréchet mean, which frequently does not result in a unique mean [71–74]. We introduce the concept of the *topological mean* for networks, defined as the minimizer according to the Wasserstein distance, mirroring how the sample mean minimizes the Euclidean distance. The squared Wasserstein distance is translation invariant such that

$$\mathcal{D}(c + \mathcal{X}_1, c + \mathcal{X}_2) = \mathcal{D}(\mathcal{X}_1, \mathcal{X}_2).$$

If we scale connectivity matrices by $c$, we have

$$\mathcal{D}(c\mathcal{X}_1, c\mathcal{X}_2) = c^2 \mathcal{D}(\mathcal{X}_1, \mathcal{X}_2).$$

**Definition 1**. *The topological mean $\mathbb{E}\mathcal{X}$ of networks $\mathcal{X}_1, \cdots, \mathcal{X}_n$ is the graph given by*

$$\mathbb{E}\mathcal{X} = \arg\min_X \sum_{k=1}^{n} \mathcal{D}(X, \mathcal{X}_k). \tag{10}$$

Unlike the sample mean, we can have many different networks with identical topology that give the minimum. Similarly, we can define the *topological variance* $\mathbb{V}\mathcal{X}$ as follows.

**Definition 2**. *The topological variance $\mathbb{V}\mathcal{X}$ of networks $\mathcal{X}_1, \cdots, \mathcal{X}_n$ is the graph given by*

$$\mathbb{V}\mathcal{X} = \frac{1}{n}\sum_{k=1}^{n}\mathcal{D}(\mathbb{E}\mathcal{X}, \mathcal{X}_k). \tag{11}$$

The topological variance can be interpreted as the variability of graphs from the topological mean $\mathbb{E}\mathcal{X}$. To compute the topological mean and variance, we only need to identify a network with identical topology as the topological mean or the topological variance.

**Theorem 3**. *Consider graphs $\mathcal{X}_i = (V, w^i)$ with corresponding birth-death decompositions $W_i = W_{ib} \cup W_{id}$ with birth sets $W_{ib} = \{b^i_{(1)}, \cdots, b^i_{(q_0)}\}$ and death sets $W_{id} = \{d^i_{(1)}, \cdots, d^i_{(q_1)}\}$. Then, there exists topological mean $\mathbb{E}\mathcal{X} = (V, w)$ with birth-death decomposition $W_b \cup W_d$ with $W_b = \{b_1, \cdots, b_{q_0}\}$ and $W_d = \{d_1, \cdots, d_{q_1}\}$ satisfying*

$$b_j = \frac{1}{n}\sum_{i=1}^{n}b^i_{(j)}, \quad d_j = \frac{1}{n}\sum_{i=1}^{n}d^i_{(j)}. \tag{12}$$

## 2.6 Topological clustering

There are few studies that used the Wasserstein distance for clustering [38, 75]. The existing methods are mainly applied to geometric data without topological consideration or persistence. It is not obvious how to apply such geometric methods to cluster graph or network data. We propose to use the Wasserstein distance to cluster collection of graphs $\mathcal{X}_1, \cdots, \mathcal{X}_n$ into $k$ clusters $C_1, \cdots, C_k$ such that

$$\cup_{i=1}^{k}C_i = \{\mathcal{X}_1, \cdots, \mathcal{X}_n\}, \quad C_i \cap C_j = \emptyset.$$

Let $C = (C_1, \cdots, C_k)$ be the collection of clusters. Let $\mu_j$ be the *topological cluster mean* within $C_j$ given by

$$\mu_j = \arg\min_{X}\sum_{\mathcal{X}_k \in C_j}\mathcal{D}(X, \mathcal{X}_k).$$

The cluster mean is computed through Theorem 3. Just like Fréchet mean, the cluster mean is not unique in a geometric sense but only unique in a topological sense [71–74]. Let $\mu = (\mu_1, \cdots, \mu_k)$ be the cluster mean vector. The within-cluster distance from the cluster centers is given by

$$l_W(C; \mu) = \sum_{j=1}^{k}\sum_{X \in C_j}\mathcal{D}(X, \mu_j). \tag{13}$$

If we let $|C_j|$ to be the number of networks within cluster $C_j$, (13) can be written as

$$l_W(C; \mu) = \sum_{j=1}^{k}|C_j|\mathbb{V}_j\mathcal{X}, \tag{14}$$

with topological cluster variance

$$\mathbb{V}_j\mathcal{X} = \frac{1}{|C_j|}\sum_{X \in C_j}\mathcal{D}(X, \mu_j)$$

within cluster $C_j$. The optimal cluster is found by minimizing within-cluster distance $l_W(C; \mu)$ in (13) over every possible partition of $C$.

If $\mu$ is given and fixed, the identification of clusters $C$ can be done easily by assigning each network to the closest mean. Thus the topological clustering algorithm can be written as the two-step optimization similar to the expectation maximization (EM) algorithm often used in variational inferences and likelihood methods [76]. The first step computes the cluster mean. The second step minimizes the within-cluster distance. Just like $k$-means clustering, the two-step optimization is then iterated until convergence. Such process converges locally.

**Theorem 4**. *The topological clustering converges locally.*

The direct algebraic proof is fairly involving and given in [17]. Here we provide a more intuitive explanation. Note $D_{W0}$ and $D_{W1}$ are Euclidean distances in convex set $\mathcal{T}_0 \otimes \mathcal{T}_1$. Subsequently,

$$\mathcal{D}(P_1, P_2) = D_{W0}^2(P_1, P_2) + D_{W1}^2(P_1, P_2).$$

is the Euclidean distance in the Cartesian product $\mathcal{T}_0 \otimes \mathcal{T}_1$. Thus, our topological clustering is equivalent to $k$-means clustering restricted to the convex set $\mathcal{T}_0 \otimes \mathcal{T}_1$. The convergence of topological clustering is then the direct consequence of the convergence of $k$-means clustering, which always converges in such a convex space. Numerically we minimize (13) by replacing the Wasserstein distance with the 2-norm between sorted vectors of birth and death values in $k$-means clustering.

Like $k$-means clustering algorithm that only converges to local minimum, there is no guarantee the topological clustering converges to the global minimum [77]. This is remedied by repeating the algorithm multiple times with different random seeds and taking the smallest possible minimum. The method is implemented as the Matlab function `WS_cluster.m` which inputs the collection of networks and outputs the cluster labels and clustering accuracy. Different choice of initial cluster centers may lead to different results. Thus, the algorithm may become stuck in a local minimum and may not converge to the global minimum. Thus, in actual numerical implementation, we used different initializations of centers. Then, we picked the best clustering result with the smallest within cluster distance $l_W$.

## 2.7 Validation

We validated the topological clustering in a simulation with the ground truth against $k$-means and hierarchical clustering [18]. We generated 4 circular patterns of identical topology (Fig 6-top) and different topology (Fig 6-bottom). Along the circles, we uniformly sampled 60 nodes and added Gaussian noise $N(0, 0.3^2)$ on the coordinates. We generated 5 random networks per group. The Euclidean distance ($L_2$-norm) between randomly generated points is used to build connectivity matrices for $k$-means and hierarchical clustering. Fig 6 shows the superposition of nodes from 20 networks. For $k$-means and Wasserstein graph clustering, the average result of 100 random seeds is reported.

**2.7.1 Testing for false positives.** In the experiment depicted in Fig 6, we evaluated the occurrence of false positives in scenarios devoid of topological differences. All groups, derived from Group 1 through rotations, are topologically identical. Hence, any detected differences are false positives. While $k$-means clustering exhibited an accuracy of 0.90 ± 0.15, and hierarchical clustering achieved perfect accuracy (1.00), these methods reported significant false positives, erroneously categorizing the groups as distinct clusters. The absence of inherent topological differences between the groups implies that higher clustering accuracy is indicative of false positive results. Conversely, topological clustering, with a lower accuracy of

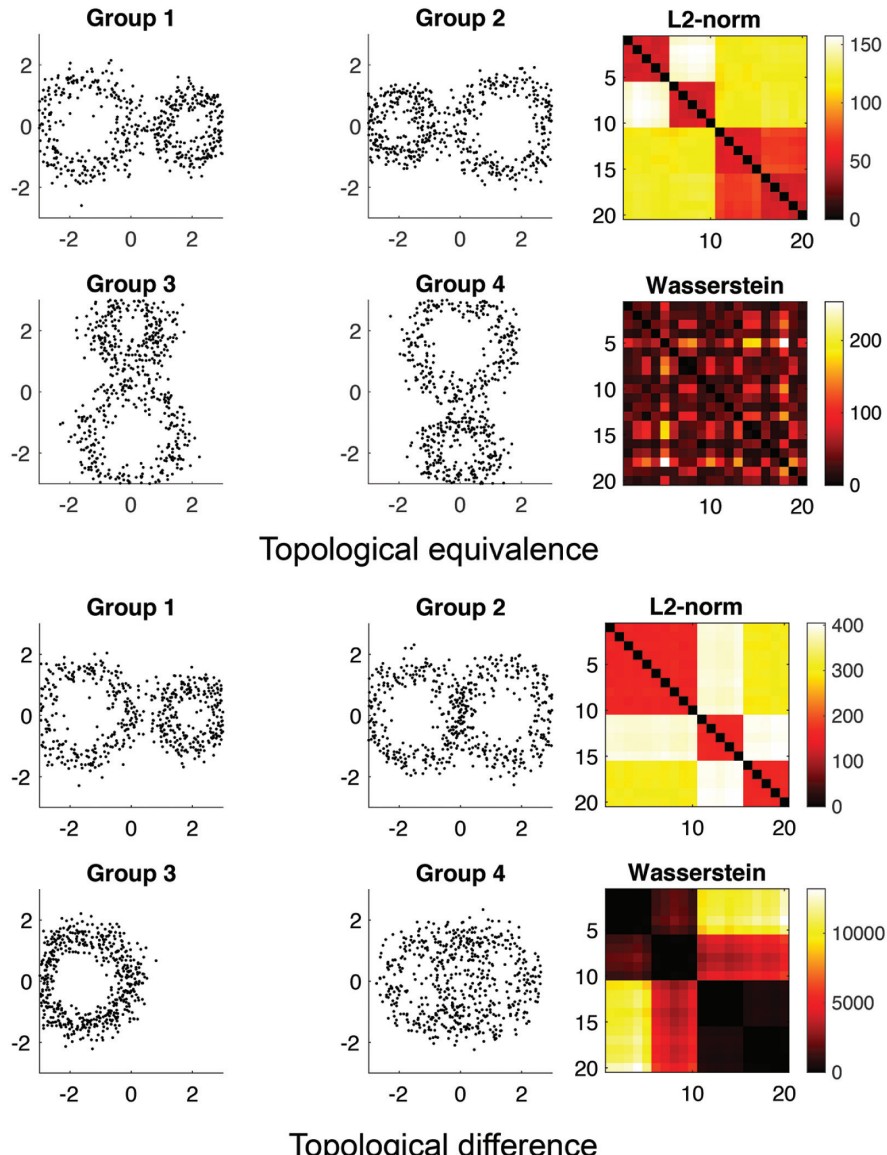

**Fig 6.** Top: simulation study on topological equivalence. The correct clustering method should *not* be able to cluster them because they are all topologically equivalent. The pairwise Euclidean distance ($L_2$-norm) is used in $k$-means and hierarchical clustering. The Wasserstein distance is used in topological clustering. Bottom: simulation study on topological difference. The correct clustering method should be able to cluster them because they are all topologically different.

$0.53 \pm 0.08$, demonstrated a reduced tendency for reporting false positives in the absence of topological differences.

**2.7.2 Testing for false negatives.** Fig 6 presents our test for false negatives, featuring groups with varying numbers of cycles and distinct topologies. In this scenario, topological differences should be detectable. Here, $k$-means clustering recorded an accuracy of $0.83 \pm 0.16$, and hierarchical clustering again reported perfect accuracy. Notably, topological clustering attained a high accuracy of $0.98 \pm 0.09$. Separating topological from geometric signals is

challenging; the presence of topological differences often coincides with geometric variations, which can influence the performance of all tested methods.

In summary, while traditional clustering methods based on geometric distances are prone to a significant number of false positives, making them less suitable for topological learning tasks, the proposed Wasserstein distance-based approach demonstrates superior performance. This method excels in minimizing both false positives and false negatives, as evidenced by our tests. Its effectiveness is particularly noteworthy in topological learning tasks, where discerning topological rather than geometric distinctions is crucial.

## 2.8 Weighted Fourier series representation

The predominant method for computing time-varying correlation in time series data, particularly in neuroimaging studies, involves Sliding Windows (SW). This technique entails computing correlations between brain regions across various time windows [77–81]. However, the use of discrete windows in SW can lead to artificially high-frequency fluctuations in dynamic correlations [82]. While tapering methods can occasionally mitigate these effects [78], the correlation computations within these windows remain susceptible to the influence of outliers [83].

To circumvent these limitations, we employed the Weighted Fourier Series (WFS) representation [84, 85]. This approach extends the traditional cosine Fourier transform by incorporating an additional exponential weight. This modification effectively smooths out high-frequency noise and diminishes the Gibbs phenomenon [84, 86]. Crucially, WFS eliminates the need for sliding windows (SW) when computing time-correlated data. Given the necessity for robust signal denoising methods to ensure the efficacy of the persistent homology method across various subjects and time points, such an approach is needed. Consider an arbitrary noise signal $f(t)$, $t \in [0, 1]$, which will undergo denoising through the diffusion process.

**Theorem 5**. *The unique solution to 1D heat diffusion:*

$$\frac{\partial}{\partial s} h(t, s) = \frac{\partial^2}{\partial t^2} h(t, s) \tag{15}$$

*on unit interval* [0, 1] *with initial condition* $h(t, s = 0) = f(t)$ *is given by WFS:*

$$h(t, s) = \sum_{l=0}^{\infty} e^{-l^2 \pi^2 s} c_{fl} \psi_l(t), \tag{16}$$

*where* $\psi_0(t) = 1$, $\psi_l(t) = \sqrt{2} \cos(l\pi t)$ *are the cosine basis and* $c_{fl} = \int_0^1 f(t) \psi_l(t) dt$ *are the expansion coefficients.*

The algebraic derivation is given in [84]. Note the cosine basis is orthonormal

$$\langle \psi_l, \psi_m \rangle = \int_0^1 \psi_l(t) \psi_m(t) \, dt = \delta_{lm},$$

where $\delta_{lm}$ is Kroneker-detal taking value 1 if $l = m$ and 0 otherwise. We can rewrite (16) as a more convenient convolution form

$$h(t, s) = \int_0^1 K_s(t, t') f(t') dt',$$

where heat kernel $K_s(t, t')$ is given by

$$K_s(t, t') = \sum_{l=0}^{\infty} e^{-l^2 \pi^2 s} \psi_l(t) \psi_l(t').$$

The diffusion time $s$ is usually referred to as the kernel bandwidth and controls the amount of smoothing. Heat kernel satisfies $\int_0^1 K_s(t, t') \, dt = 1$ for any $t'$ and $s$.

To reduce unwanted boundary effects near the data boundary $t = 0$ and $t = 1$ [77, 86], we project the data onto the circle $\mathcal{C}$ with circumference 2 by the mirror reflection:

$$g(t) = f(t) \text{ if } t \in [0, 1], \quad g(t) = f(2 - t) \text{ if } t \in [1, 2].$$

Then perform WFS on the circle.

**Theorem 6**. *The unique solution to 1D heat diffusion:*

$$\frac{\partial}{\partial s} h(t, s) = \frac{\partial^2}{\partial t^2} h(t, s) \tag{17}$$

*on the circle $\mathcal{C}$ with the initial periodic condition $h(t, s = 0) = f(t)$ if $t \in [0, 1]$, $h(t, s = 0) = f(2 - t)$ if $t \in [1, 2]$ is given by WFS:*

$$h(t, s) = \sum_{l=0}^{\infty} e^{-l^2 \pi^2 s} c_{fl} \psi_l(t), \tag{18}$$

*where $\psi_0(t) = 1$, $\psi_l(t) = \sqrt{2} \cos(l\pi t)$ are the cosine basis and $c_{fl} = \int_0^1 f(t) \psi_l(t) dt$ are the expansion coefficients.*

The cosine series coefficients $c_{fl}$ are estimated using the least squares method by setting up a matrix equation [84]. We set the expansion degree to equate the number of time points, which is 295. The window size of 20 TRs was used in most sliding window methods [77, 78, 87]. We matched the full width at half maximum (FWHM) of heat kernel to the window size numerically. We used the fact that diffusion time $s$ in heat kernel approximately matches to the kernel bandwidth of Gaussian kernel $e^{-t^2/2\sigma^2}$ as $\sigma = s^2/2$ (page 144 in [88]). 20 TRs is approximately equivalent to heat kernel bandwidth of about $4.144 \cdot 10^{-4}$ in terms of FWHM. Fig 7 displays the WFS representation of rsfMRI with different kernel bandwidths.

## 2.9 Dynamic correlation on weighted Fourier series

The weighted Fourier series representation provides a way to compute correlations dynamically without using sliding windows. Consider time series $x(t)$ and $y(t)$ with heat kernel $K_s(t, t')$. The mean and variance of signals with respect to the heat kernel are given by

$$\mathbb{E}x(t) = \int_0^1 K_s(t, t') x(t') \, dt',$$

$$\mathbb{V}x(t) = \int_0^1 K_s(t, t') x^2(t') \, dt' - [\mathbb{E}x(t)]^2.$$

3 Subsequently, the correlation $w(t)$ of $x(t)$ and $y(t)$ is given by

$$w(t) = \frac{\int_0^1 K_s(t, t') x(t') y(t') \, dt' - \mathbb{E}x(t)\mathbb{E}y(t)}{\sqrt{\mathbb{V}x(t)} \sqrt{\mathbb{V}y(t)}}. \tag{19}$$

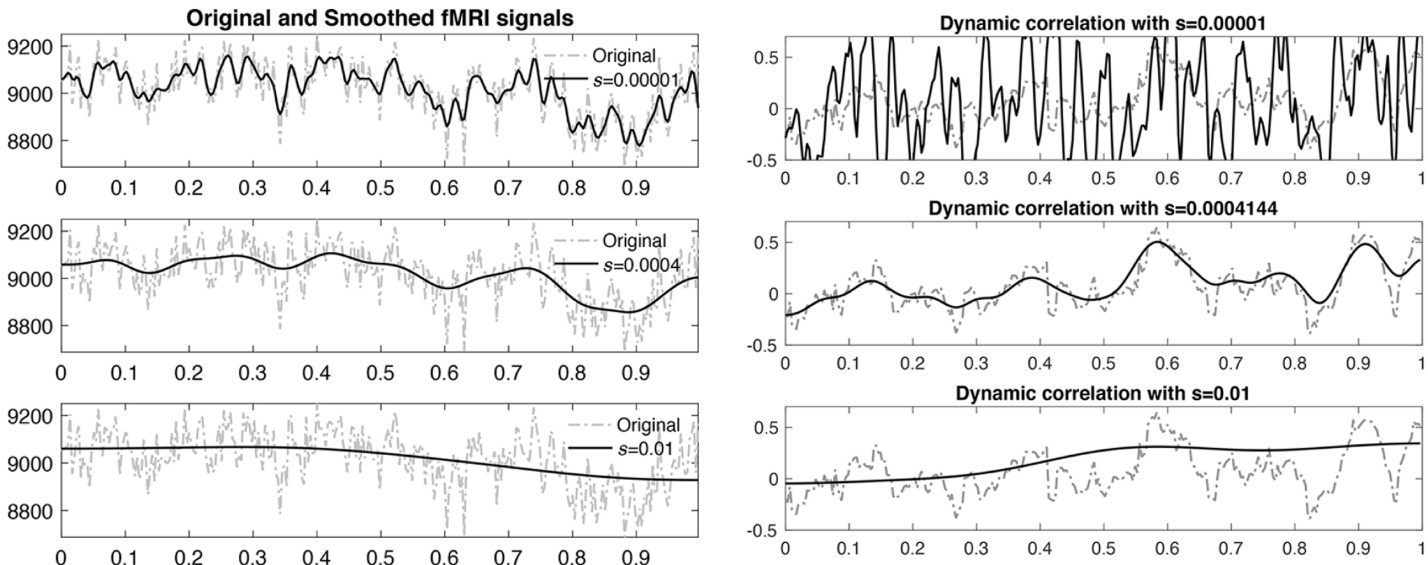

**Fig 7.** Left: The original and smoothed fMRI time series using WFS with degree $L = 295$ and different heat kernel bandwidth $s$. The bandwidth $4.141 \times 10^{-4}$ used in this study approximately matches 20 TRs often used in the sliding window methods. Right: The dotted gray lines are correlations computed over sliding windows. The solid black lines are correlations computed using WFS.

When the kernel is shaped as a sliding window, the correlation $w(t)$ exactly matches the correlation computed over the sliding window. The kernelized correlation generalizes the concept of integral correlations with the additional weighting term [89]. As $s \to \infty$, $w(t)$ converges to the Pearson correlation computed over the whole time points. Thus, the kernel bandwidth behaves like the length of sliding window.

**Theorem 7.** *The correlation $w(t)$ of time series $x(t)$ and $y(t)$ with respect to heat kernel $K_s(t, t')$ is given by*

$$w(t) = \frac{\sum_{l=0}^{\infty} e^{-l^2\pi^2 s} c_{xyl}\psi_l(t) - \mu_x(t)\mu_y(t)}{\sigma_x(t)\sigma_y(t)}, \tag{20}$$

*with*

$$\mu_x(t) = \sum_{l=0}^{\infty} e^{-l^2\pi^2 s} c_{xl}\psi_l(t), \quad \sigma_x^2(t) = \sum_{l=0}^{\infty} e^{-l^2\pi^2 s} c_{xxl}\psi_l(t) - \mu_x^2(t).$$

$$c_{xl} = \int_0^1 x(t)\psi_l(t)dt, \quad c_{yl} = \int_0^1 y(t)\psi_l(t)dt$$

*are the cosine series coefficients. Similarly we expand $x(t)y(t)$, $x^2(t)$ and $y^2(t)$ using the cosine basis and obtain coefficients $c_{xyl}$, $c_{xxl}$ and $c_{yyl}$.*

The derivation follows by simply replacing all the terms with the WFS representation. Correlation (20) is the formula we used to compute the dynamic correlation in this study. Fig 7 displays the WFS-based dynamic correlation for different bandwidths. A similar weighted correlation was proposed in [90], where the time varying exponential weights proportional to $e^{t/\theta}$ with exponential decay factor $\theta$ were used. However, our exponential weight term is related to the spectral decomposition of heat kernel in the spectral domain and invariant over time. The WFS based correlation is not related to [90].

## 3 Results

### 3.1 Data

The proposed method is applied in the accurate estimation of state spaces in dynamically changing functional brain networks. The 479 subjects resting-state functional magnetic resonance images (rs-fMRI) used in this paper were collected on a 3T MRI scanner (Discovery MR750, General Electric Medical Systems, Milwaukee, WI, USA) with a 32-channel RF head coil array. The 479 healthy subjects consist of 231 males and 248 females ranging in age from 13 to 25 years. The sample contains 132 monozygotic (MZ) twin pairs and 93 same-sex dizygotic (DZ) twin pairs.

The image preprocessing includes motion corrections and image alignment to the MNI template and follows [91, 92]. The resulting rs-fMRI consist of $91 \times 109 \times 91$ isotropic voxels at 295 time points. We further parcellated the brain volume into 116 non-overlapping brain regions using the Automated Anatomical Labelling (AAL) atlas [93]. The fMRI data were averaged across voxels within each brain region, resulting in 116 average fMRI signals with 295 time points for each subject. The rs-fMRI signals were then scaled to fit to unit interval [0, 1] and treated as functional data in [0, 1].

### 3.2 Topological state space estimation

For $p$ brain regions, we estimated $p \times p$ dynamically changing correlation matrices $C_i(t)$ for the $i$-th subject using WFS. Let $\mathbf{C}_{ij}$ denote the vectorization of the upper triangle of $p \times p$ matrix $C_i(t_j)$ at time point $t_j$ into $p^2 \times 1$ vector. For each fixed $i$, the collection of $\mathbf{C}_{ij}$ over $T = 295$ time points is then feed into topological clustering in identifying the recurring brain connectivity at the subject level. We are clustering individual brain networks without putting any constraint on group or twin. We compared the proposed Wasserstein clustering against the $k$-means clustering, which has been often used as the baseline method in the state space modeling [77, 78, 86]. After clustering, each correlation matrix $C_i(t_j)$ is assigned integers between 1 and $k$. These discrete states serve as the basis for investigating the dynamic pattern of brain connectivity [94]. For the convergence of both topological clustering and $k$-means clustering, the clusterings were repeated 10 times with different initial centroids and the best result (smallest within-cluster distance) is reported. Fig 8-left displays the result of the topological clustering against the $k$-means for three subjects. 295 time points are rescaled to fit into unit interval [0, 1].

The optimal number of cluster $k$ was determined by the *elbow method* [77, 78, 94, 95]. For each value of $k$, we computed the ratio of the within-cluster to between-cluster distances. The ratio shows the goodness-of-fit of the cluster model. The elbow method gives the largest slope change in the ratio when $k = 3$ in the both methods (Fig 8-right). At $k = 3$, the ratio is $0.034 \pm 0.012$ for 479 subjects for Wasserstein while it is $0.202 \pm 0.047$ for the $k$-means. The *six* times smaller ratio for the topological clustering demonstrates the superior model fit over $k$-means. Fig 9 shows the results of clustering for both methods. The $k$-means clustering result is based on averaging correlations of every time point and subject within each state. The resulting states in the $k$-means clustering are somewhat random without any biologically interpretable pattern. The topological clustering computes the *topological mean* of every time point and subject within each state.

### 3.3 Twin correlations over transpositions

Using additional twin information in the data, we further investigated if the state change pattern itself is genetically heritable. As far as we are aware, there is no study on the heritability of the state change pattern itself. This requires computing twin correlations. We assume there are

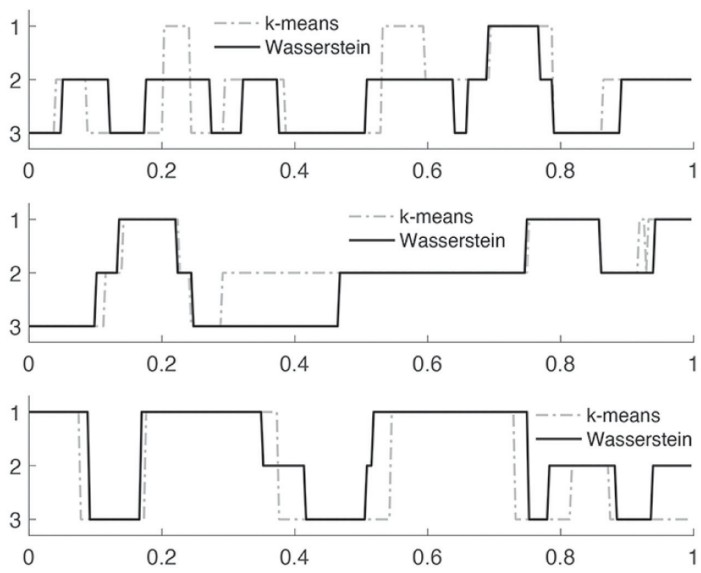 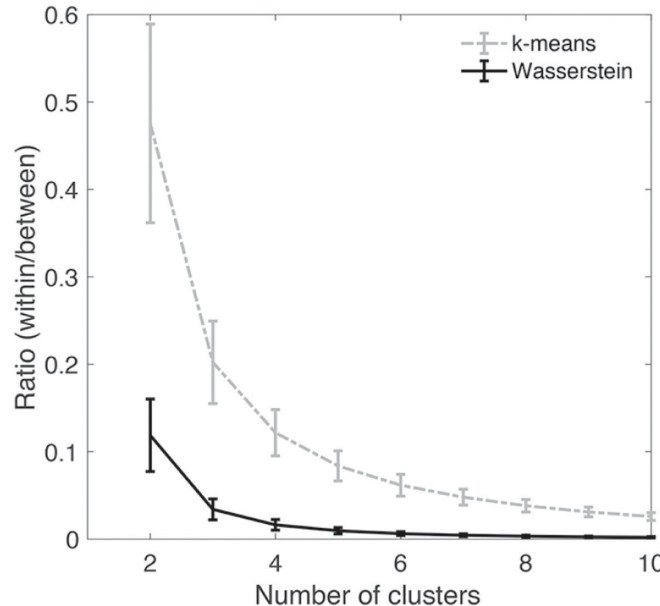

**Fig 8.** Left: The time series of estimated state spaces using the topological clustering and *k*-means clustering for 3 subjects. The time is normalized into unit interval [0, 1]. Right: The ratio of within-cluster to between-cluster distances. Smaller the ratio, better the clustering fit is.

*m* MZ- and *n* DZ-twins. For some feature, let $x_i = (x_{i1}, x_{i2})^\top$ be the *i*-th twin pair in MZ-twin and $y_i = (y_{i1}, y_{i2})^\top$ be the *i*-th twin pair in DZ-twin. They are represented as

$$\mathbf{x} = \begin{pmatrix} x_{11}, & \cdots & , x_{m1} \\ x_{12}, & \cdots & , x_{m2} \end{pmatrix}, \quad \mathbf{y} = \begin{pmatrix} y_{11}, & \cdots & , y_{n1} \\ y_{12}, & \cdots & , y_{n2} \end{pmatrix}.$$

Let $\mathbf{x}_j$ be the *j*-th row of $\mathbf{x}$, i.e., $\mathbf{x}_j = (x_{1j}, x_{2j}, \cdots, x_{mj})$. Similarly let $\mathbf{y}_j = (y_{1j}, y_{2j}, \cdots, y_{nj})$. Then MZ- and DZ-correlations are computed as the sample correlation

$$\begin{aligned} \gamma^{MZ}(\mathbf{x}_1, \mathbf{x}_2) &= corr(\mathbf{x}_1, \mathbf{x}_2) \\ \gamma^{DZ}(\mathbf{y}_1, \mathbf{y}_2) &= corr(\mathbf{y}_1, \mathbf{y}_2). \end{aligned}$$

However, there is no preference for the order of twins within a pair, and we can *transpose* the *i*-th twin pair in MZ-twin such that

$$\begin{aligned} \tau_i(\mathbf{x}_1) &= (x_{11} \cdots x_{i-1,1}, x_{i2}, x_{i+1,1} \cdots x_{m1}), \\ \tau_i(\mathbf{x}_2) &= (x_{12} \cdots x_{i-1,2}, x_{i1}, x_{i+1,2} \cdots x_{m2}) \end{aligned}$$

and obtain another twin correlation $\gamma^{MZ}(\tau_i(\mathbf{x}_1), \tau_i(\mathbf{x}_2))$ [96, 97]. Ignoring symmetry, there are $2^m$ possible combinations in ordering the twins, which form a permutation group. The size of the permutation group grows exponentially large as the sample size increases. Computing correlations over all permutations is not even computationally feasible for large *m* beyond 100. Fig 10 illustrates many possible transpositions within twins. Thus, we propose a new fast online computational strategy for computing twin correlations.

Over transposition $\tau_i$, the correlation changes

$$\gamma^{MZ}(\mathbf{x}_1, \mathbf{x}_2) \rightarrow \gamma^{MZ}(\tau_i(\mathbf{x}_1), \tau_i(\mathbf{x}_2)) \tag{21}$$

incrementally. We will determine the exact increment over the transposition. The sample

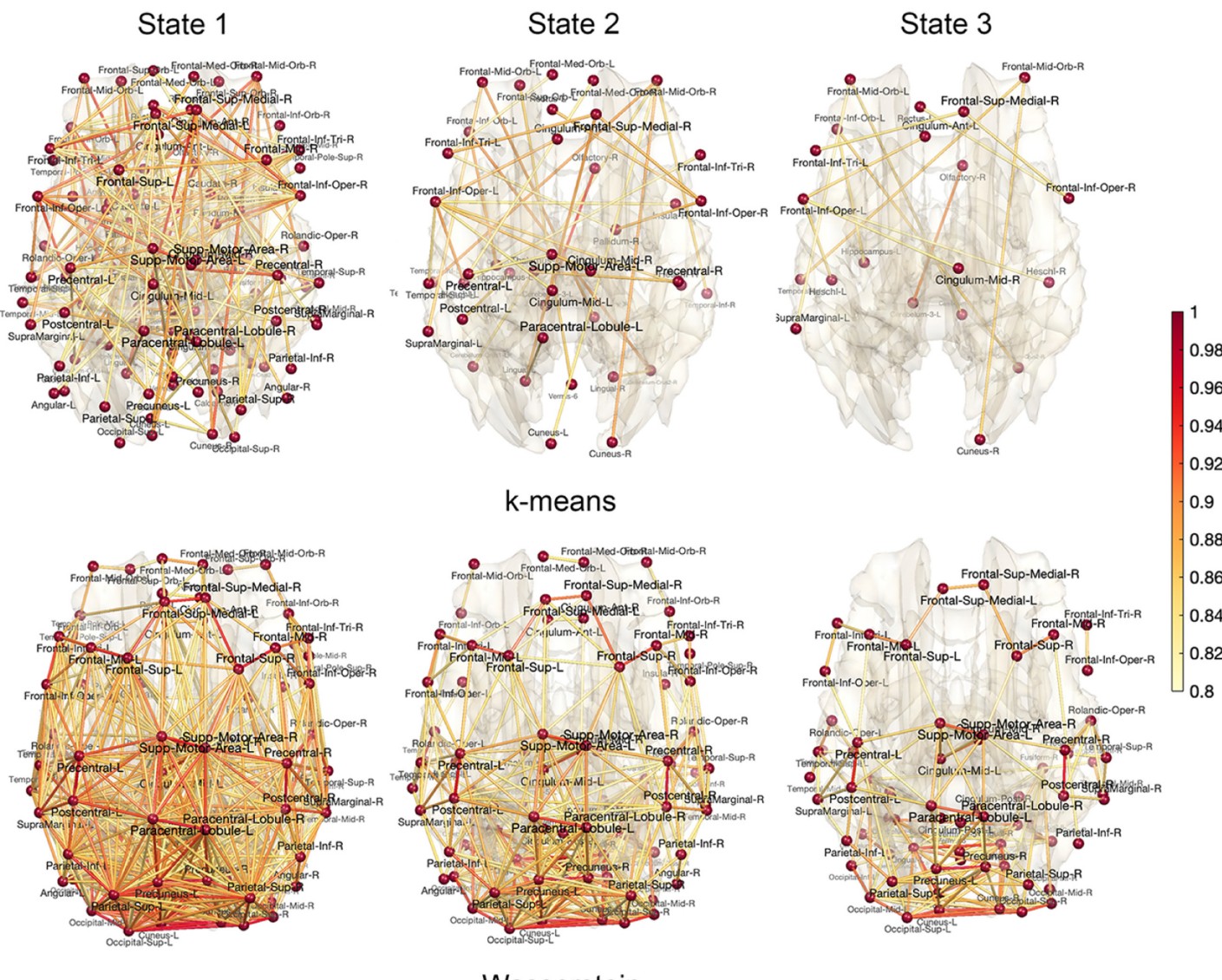

**Fig 9. The estimated state spaces of dynamically changing brain networks.** The correlations are averaged over every time point and subject within each state for *k*-means clustering (top) and Wasserstein distance based topological clustering (bottom). In *k*-means clustering, the connectivity pattern of each state is somewhat random. In topological clustering, the connectivity pattern is highly symmetric even though we did not put any symmetry constraint in the clustering method.

correlation between $\mathbf{x}_k$ and $\mathbf{x}_l$ involves the following functions.

$$v(\mathbf{x}_k) = \sum_{l=1}^{m} x_{lk}$$

$$\omega(\mathbf{x}_k, \mathbf{x}_l) = \sum_{r=1}^{m} (x_{rk} - v(\mathbf{x}_k)/m)(x_{rl} - v(\mathbf{x}_l)/m).$$

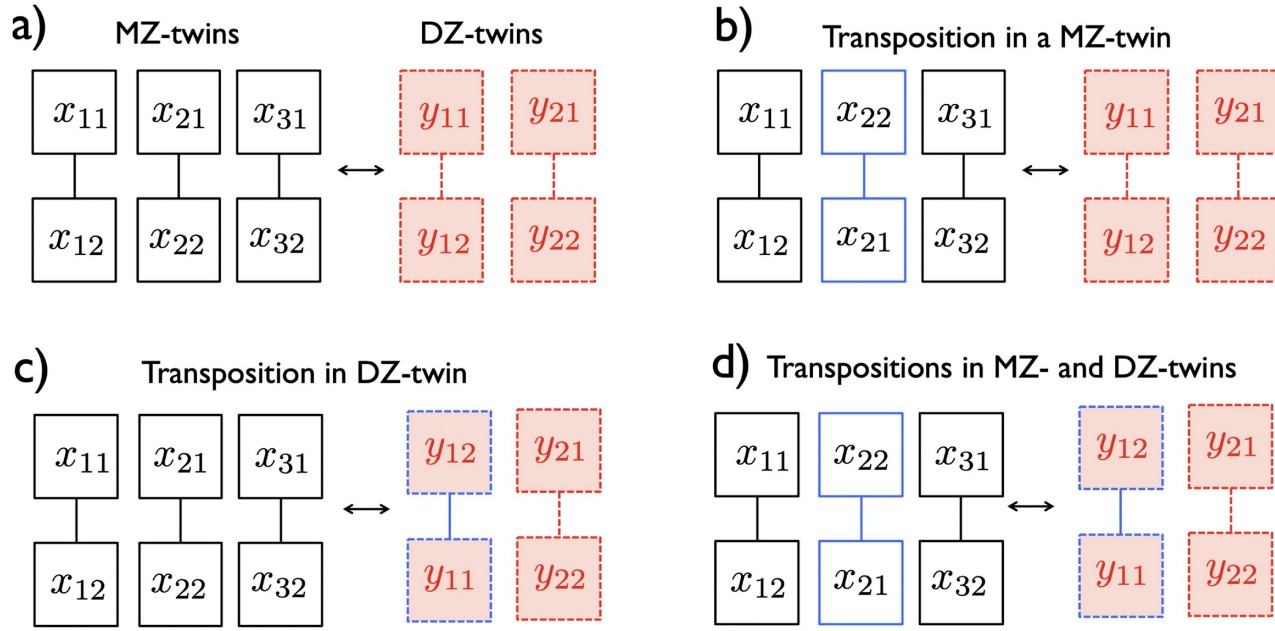

**Fig 10. The schematic of transpositions on 3 MZ- and 2 DZ-twins.** a) One possible pairing. b) Transposition within a MZ-twin. c) Transposition within a DZ-twin. d) Transpositions in both MZ- and DZ-twins simultaneously. Any transposition will affect the heritability estimate so it is necessary to account for as many transpositions as possible.

The functions $\mu$ and $\omega$ are updated over transposition $\tau_i$ as

$$v(\tau_i(\mathbf{x}_k)) = v(\mathbf{x}_k) - x_{ik} + x_{il}$$
$$\omega(\tau_i(\mathbf{x}_k), \tau_i(\mathbf{x}_l)) = \omega(\mathbf{x}_k, \mathbf{x}_l) + (x_{ik} - x_{il})^2/m - (x_{ik} - x_{il})(v(\mathbf{x}_k) - v(\mathbf{x}_l))/m.$$

Then the MZ-twin correlation after transposition is updated as

$$\gamma^{MZ}(\tau_i(\mathbf{x}_1), \tau_i(\mathbf{x}_2)) = \frac{\omega(\tau_i(\mathbf{x}_1), \tau_i(\mathbf{x}_2))}{\sqrt{\omega(\tau_i(\mathbf{x}_1), \tau_i(\mathbf{x}_1))\omega(\tau_i(\mathbf{x}_2), \tau_i(\mathbf{x}_2))}}. \tag{22}$$

The time complexity for correlation computation is 33 operations per transposition, which is substantially lower than the computational complexity of directly computing correlations per permutation. In the numerical implementation, we sequentially apply random transpositions $\tau_{i_1}, \tau_{i_2}, \cdots, \tau_{i_J}$. This results in $J$ different twin correlations, which are averaged. Let

$$\pi_1 = \tau_{i_1}, \pi_2 = \tau_{i_2} \circ \tau_{i_1}, \cdots, \pi_J = \tau_{i_J} \circ \cdots \circ \tau_{i_2} \circ \tau_{i_1}.$$

The average correlation $\bar{\gamma}_J^{MZ}$ of all $J$ transpositions is given by

$$\bar{\gamma}_J^{MZ} = \frac{1}{J}\sum_{j=1}^{J}\gamma^{MZ}(\pi_{i_j}(\mathbf{x}_1), \pi_{i_j}(\mathbf{x}_2)).$$

In each sequential update, the average correlation can be updated iteratively as

$$\bar{\gamma}_J^{MZ} = \frac{J-1}{J}\bar{\gamma}_{J-1}^{MZ} + \frac{1}{J}\gamma^{MZ}(\pi_{i_j}(\mathbf{x}_1), \pi_{i_j}(\mathbf{x}_2)).$$

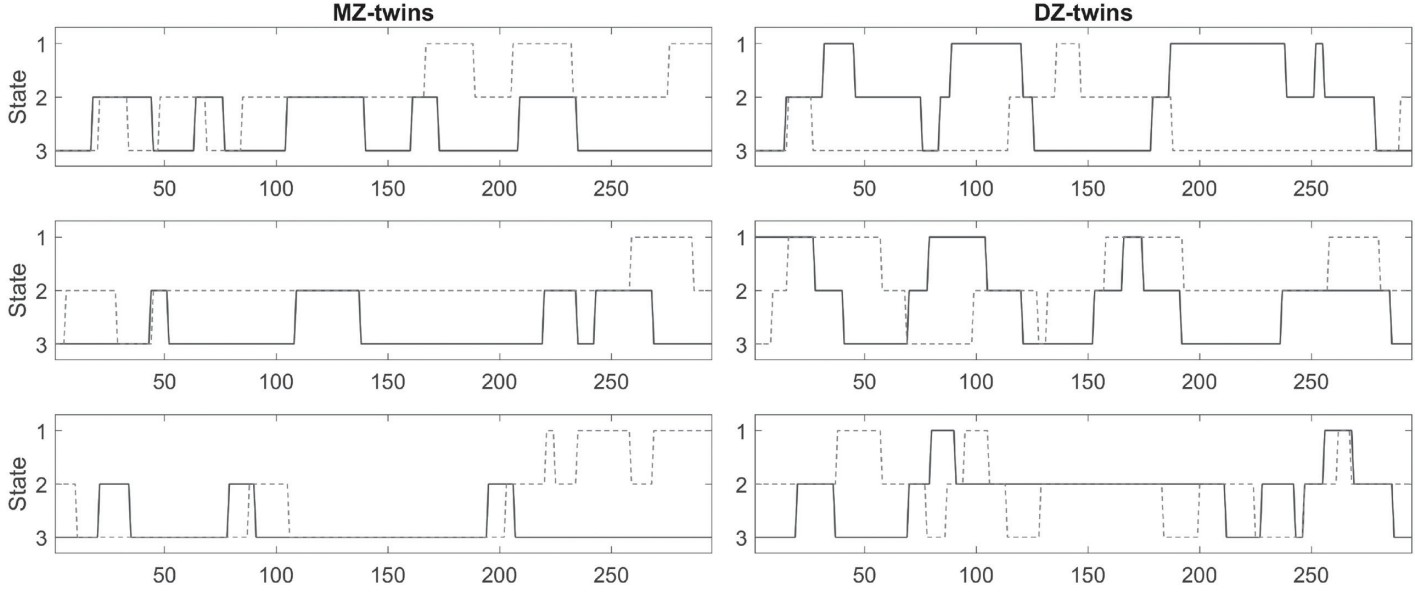

**Fig 11. State visits for 3 MZ-twins (left) and 3 DZ-twins (right) obtained from the baseline *k*-means clustering.** We are interested in determining the heritability of such state changes. Unfortunately, even within a twin, the time series of state change do not synchronize, making the task extremely challenging.

If we use enough transpositions, the average correlation $\bar{\gamma}_J^{MZ}$ converges to the true underlying twin correlation $\gamma^{MZ}$ for sufficiently large $J$. DZ-twin correlation $\gamma^{DZ}$ is estimated similarly.

In the widely used ACE genetic model, the heritability index (HI) $h$, which determines the amount of variation due to genetic difference in a population, is estimated using Falconer's formula [45, 98, 99]. MZ-twins share 100% of genes while same-sex DZ-twins share 50% of genes on average. Thus, the additive genetic factor $A$ and the common environmental factor $C$ are related as

$$\begin{aligned} \gamma^{MZ} &= A + C, \\ \gamma^{DZ} &= A/2 + C. \end{aligned}$$

HI $h$, which measures the contribution of $A$, is given by

$$h(\mathbf{x}, \mathbf{y}) = 2(\gamma^{MZ} - \gamma^{DZ}).$$

In numerical implementation, 100 million transpositions can be easily done in 100 seconds in a desktop. Similarly, we update the DZ-correlation over the transposition.

## 3.4 Heritability of the state space

The heritability estimation of state space is not a trivial task since the estimated state does not synchronize across twins making the task fairly difficult. Fig 11 displays the state visits in randomly selected 3 MZ- and 3 DZ-twins. However, the time series of state changes do not synchronize within twins. This is likely a reason for the lack of reported heritability of the state space in the literature.

For each subject, we computed the average correlation of each state, where the average is taken over all time points within each state. The correlation matrices are then used as the input to the transposition based twin correlations [98]. Subsequently, we computed the MZ- and DZ-twin correlations within each state (Fig 12). The MZ-twin correlations (Fig 12-top) are

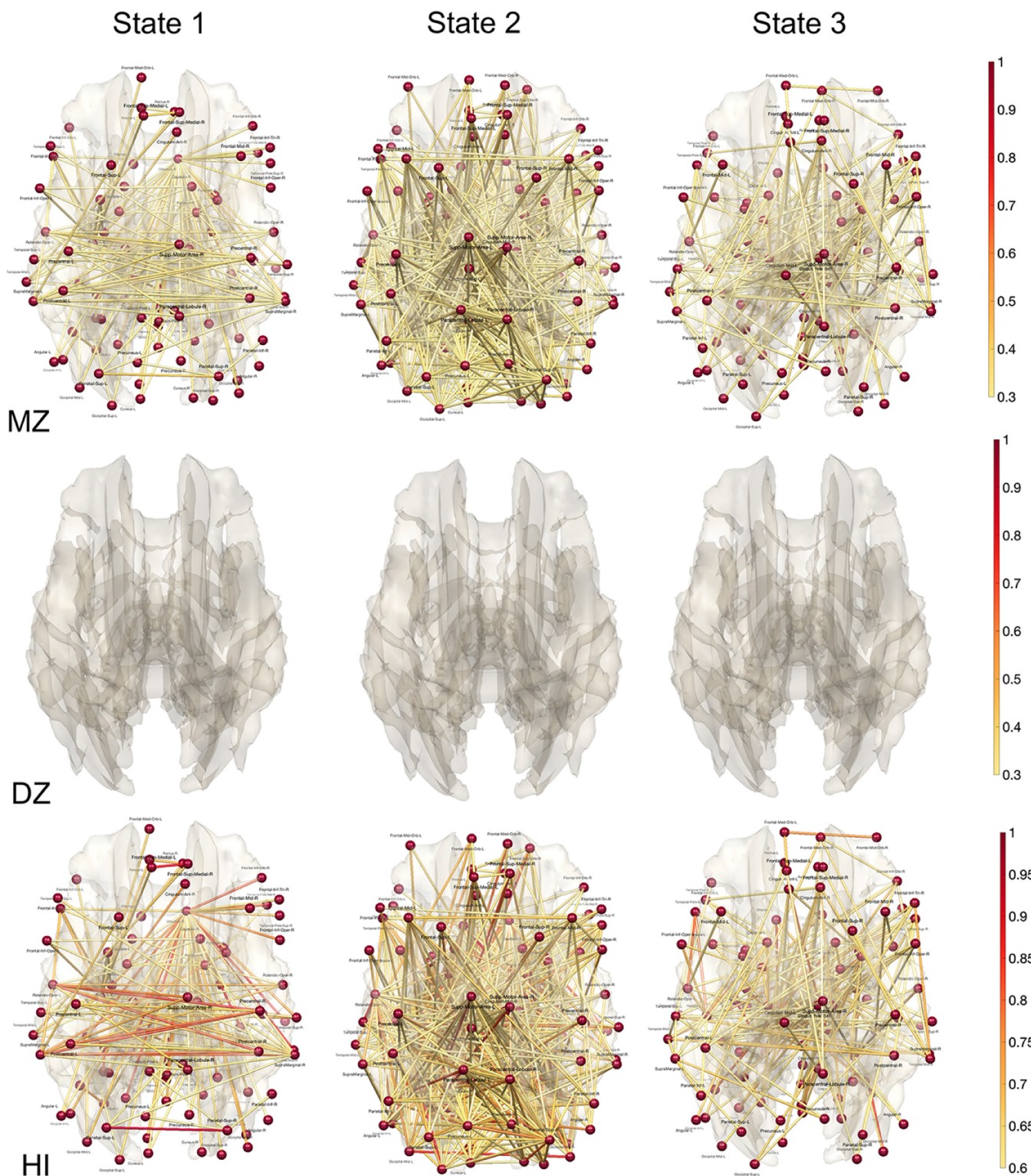

**Fig 12. MZ-correlation (top) and DZ-correlation (middle) in each state obtained through topological clustering in Fig 9.** There is no DZ-correlation above 0.3 and not displayed. The heritability index (HI) is determined by the twice the difference in twin correlations. HI of each state shows the extensive genetic contribution of dynamically changing states. The first state is characterized by prominent bilateral connections between the left and right hemispheres, whereas the second state primarily features front-back connections.

densely observed in many connections while there are no DZ-twin correlations (Fig 12-middle) observed above 0.3. We then computed the heritability index (HI) of each state (Fig 12-bottom). The heritability of the first state is characterized by strong lateralization of the hemisphere connections. The heritability of the second state is characterized by front and back connections. We believe the topological approach provides far more accurate and stable heritability index maps for dynamically changing state, which are biologically interpretable.

We reported 10 connections that give the highest HI values in all three states in Tables 1, 2 and 3. Although numerous studies report high heritability for anatomical features such as gray matter density, there are few rs-fMRI studies reporting heritability of rs-fMRI [52, 58]. Most of these studies report low HI compared to our high HI. [52] reported HI of 0.104 in the left cerebellum, 0.304 in the right cerebellum, 0.331 in the left temporal parietal region, 0.420 in the right temporal parietal region. [58] reported HI of 0.41 in the connection between the posterior cingulate cortex and right inferior parietal cortex in the default mode network involving 79 MZ- and 46 same-sex DZ-twins. Other connections are all reporting very low HI below 0.24. We believe our topological method is clustering topologically similar functional network patterns and significantly boost genetic signals.

## 3.5 Null test on twin study design

Because we are reporting significantly higher diffused heritability compared to existing literature [52, 57, 58], we performed the null test to check the validity of our analysis pipeline further. We generated the null MZ-twin data by randomly pairing each MZ individual with another, excluding their own twin. Such a permutation is generated by *derangement*, which is a permutation of the elements of a set, such that no element appears in its original position [100]. In other words, if we have a set of distinct items and you rearrange them, a derangement means none of the items are in the spot they started in. The null DZ-twin data is constructed similarly. Such null data should not show any genetic relations beyond random chances. On the null data, we recomputed the twin correlations and the heritability index, following the same pipeline as before. Fig 13 shows an example of one possible derangement out of exponentially many such permutations. For $m$ MZ-twin pairs, there are

$$m! \sum_{i=0}^{m} \frac{(-1)^i}{i!}$$

number of derangements. For the null test, we generated 1000 derangements that followed the

**Table 1. 10 connections with the highest heritability index for state 1.** Connections are sorted with respect to HI values.

| Regions | Regions | HI |
|---|---|---|
| Parietal-Sup-L | Parietal-Sup-R | 0.96 |
| Frontal-Sup-Medial-L | Frontal-Sup-Medial-R | 0.90 |
| Olfactory-R | Temporal-Mid-R | 0.89 |
| Precentral-R | Rolandic-Oper-L | 0.88 |
| Olfactory-R | Temporal-Inf-R | 0.88 |
| Olfactory-R | Fusiform-R | 0.87 |
| Olfactory-R | Cerebelum-4-5-L | 0.87 |
| Precentral-R | SupraMarginal-L | 0.85 |
| Rolandic-Oper-L | Postcentral-R | 0.84 |
| Olfactory-R | Lingual-L | 0.84 |

**Table 2. 10 connections with the highest heritability index for state 2.** Connections are sorted with respect to HI values.

| Regions | Regions | HI |
| --- | --- | --- |
| Hippocampus-L | Cerebelum-4-5-L | 1.00 |
| Olfactory-L | Fusiform-R | 0.92 |
| Precuneus-R | Cerebelum-Crus2-L | 0.90 |
| Occipital-Sup-L | Fusiform-L | 0.89 |
| Supp-Motor-Area-L | Cerebelum-Crus2-L | 0.88 |
| Occipital-Mid-L | Occipital-Mid-R | 0.87 |
| Thalamus-L | Cerebelum-9-L | 0.86 |
| Rolandic-Oper-L | Temporal-Sup-L | 0.85 |
| Paracentral-Lobule-L | Cerebelum-Crus2-L | 0.85 |
| Caudate-R | Cerebelum-Crus2-L | 0.85 |

proposed pipeline in computing average MZ- and DZ-correlations in each state. We used the Wasserstein distance in measuring the topological discrepancy. Fig 14 displays the normalized histogram of the Wasserstein distance between average MZ- and DZ-twin correlations within each state over 1000 derangements. Because the generated null data has no genetic signal, we are basically computing the Wasserstein distance between two random connectivity matrices. In comparison, the observed Wasserstein distance (red line) between average MZ- and DZ-twin correlation shows huge topological differences. None of the derangements show the large wide spread signals as our observation. We conclude that what we observe is genetic signal and cannot possibly be produced by random chance.

## 4 Discussion

In this study, we proposed the topological clustering method for the estimation and quantification of dynamic state changes in time-varying brain networks. A coherent statistical theory, grounded in persistent homology, was developed, and we demonstrated the application of this method to resting-state fMRI data. Resting-state brain networks tend to persist in a single state for extended periods before transitioning to another state [78, 80, 101, 102]. The average brain network in each state appears to diverge from the patterns reported in previous studies (Fig 10) [77, 103, 104]. Further research is required for independent validation.

**Table 3. 10 connections with the highest heritability index for state 3.** Connections are sorted with respect to HI values.

| Regions | Regions | HI |
| --- | --- | --- |
| Hippocampus-R | Cerebelum-3-R | 1.00 |
| Hippocampus-L | Cerebelum-4-5-L | 0.93 |
| Occipital-Mid-R | Cerebelum-Crus2-R | 0.86 |
| Olfactory-L | Cerebelum-3-L | 0.81 |
| Heschl-L | Temporal-Pole-Sup-L | 0.81 |
| Rolandic-Oper-L | Temporal-Pole-Sup-L | 0.80 |
| Caudate-R | Cerebelum-Crus1-L | 0.79 |
| Cerebelum-7b-R | Cerebelum-9-R | 0.78 |
| Cingulum-Ant-L | Cerebelum-3-R | 0.78 |
| Frontal-Mid-Orb-R | Frontal-Med-Orb-L | 0.78 |

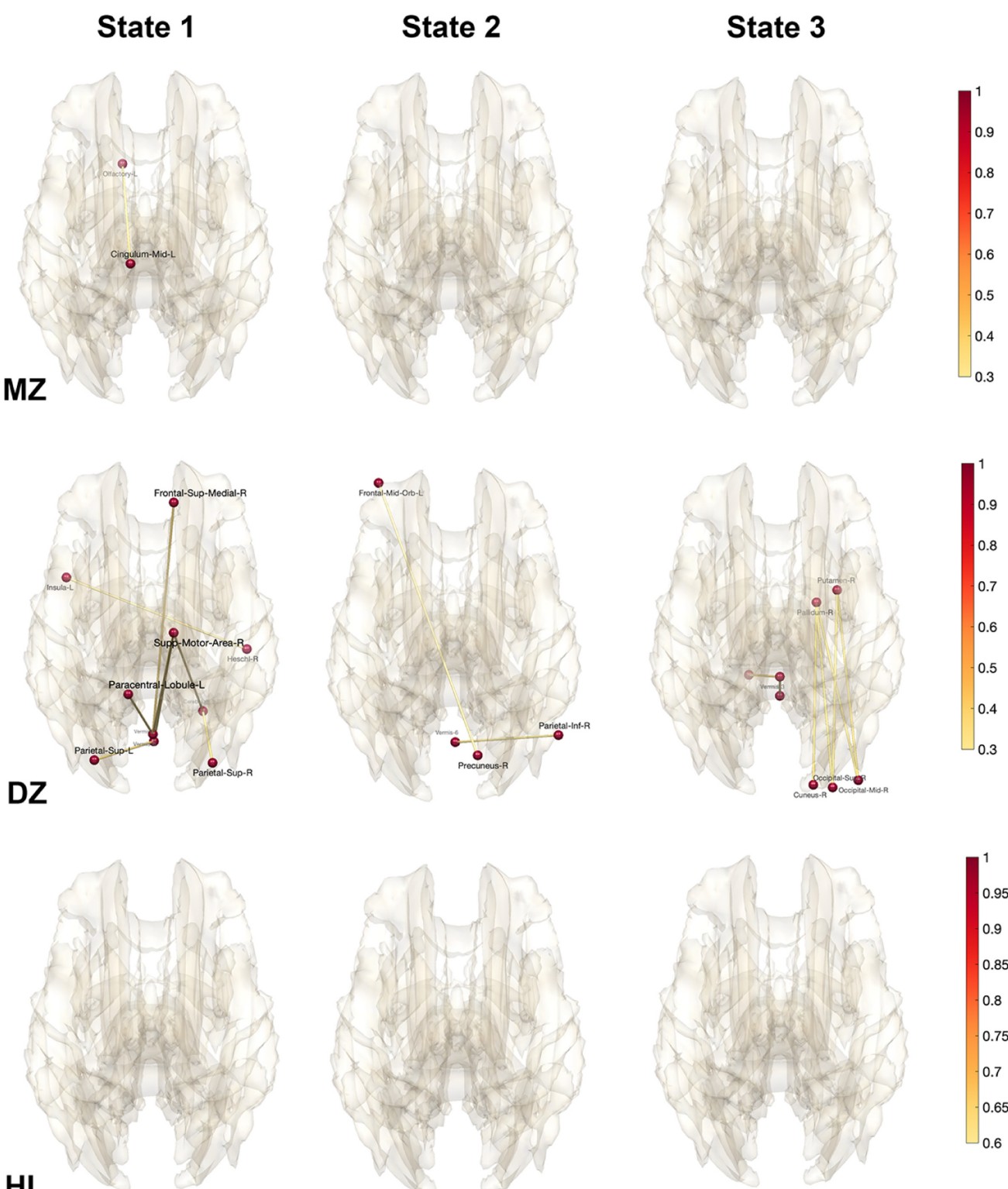

**Fig 13. MZ-correlation (top) and DZ-correlation (middle) in each state obtained through topological clustering in Fig 9.** There is no MZ-correlation above 0.3 and not displayed. The heritability index (HI) is determined by the twice the difference in twin correlations. HI of each state shows extensive genetic contribution of dynamically changing states.

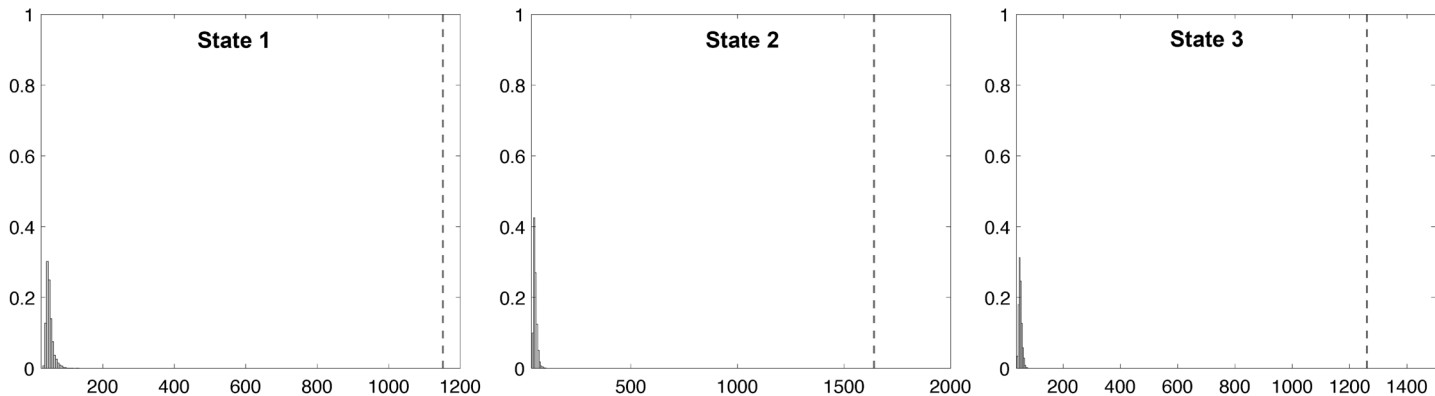

**Fig 14. The normalized histogram of the Wasserstein distance between average MZ- and DZ-twin correlations within each state over 1000 derangements.** Since the generated null data has no genetic signal, we are basically computing the Wasserstein distance between two connectivity matrices with random noises. In comparison, the observed Wasserstein distance (dotted line) between average MZ- and DZ-twin correlation shows huge topological differences.

In contrast to previous studies that reported relatively low heritability in functional brain networks [52, 57, 58, 105], our findings indicate significant higher heritability across various regions of the brain network. This discovery not only challenges the prevailing understanding but also opens new avenues for exploring genetic influences on brain network dynamics. Our observations align with the early findings by [56], which documented higher heritability in EEG spectra. Heritability in brain networks may be more nuanced than previously understood. In our framework, rather than directly using the connectivity strength, we decomposed networks into discrete topological states and computed heritability for each state. This granular analysis provides a more accurate estimation of heritability across different functional states of the brain. The resting state measures employed in studies such as those by [52, 57, 58] directly rely on *static* connectivity matrices. These matrices, while informative, often do not capture the dynamic and configural nature of brain networks. Such methods may overlook hidden configural patterns that hold significant heritable information. Our topological method represents a significant advancement in this regard. By focusing on the topological aspects of dynamic brain networks, our method is adept at identifying and extracting hidden patterns of high heritability that might be missed by traditional approaches. This capability could be crucial for understanding the genetic basis of various neuropsychiatric and neurodevelopmental disorders, where altered brain network configurations play a critical role.

Intraclass correlation (ICC) has long been recognized as a vital reliability and reproducibility metric, especially for gauging similarity in paired data when the order of pairing is not preserved [96, 106, 107]. In brain imaging, it serves as a popular baseline for test-retest (TRT) reliability assessments, often in conjunction with the Dice coefficient [108–112]. The widespread use of ICC in these contexts underscores its perceived utility in evaluating consistency across imaging sessions or different imaging modalities. The conventional computation of ICC is typically through an ANOVA statistical model, which can be fairly limited and inflexible. Recent years have seen a shift towards mixed-effects models, which offer greater flexibility and accuracy in estimating ICC, especially in datasets with nested or hierarchical structures [96, 113]. In light of these advancements, our proposed transposition-based approach for computing correlation over paired data presents a novel approach to computing ICC, potentially offering a faster and more efficient alternative. The full potential and utility of the transposition-based method for ICC computation, however, remain to be explored in future research.

## Acknowledgments

We would like to thank Chee-Ming Ting and Hernando Ombao of KAUST for discussion on $k$-means clustering. We also like to thank Soumya Das, Tananun Songdechakraiwut of University of Wisconsin, Madison and Botao Wang of University of Illinois, Urbana-Champaign for discussion on the clustering.

## Author Contributions

**Conceptualization:** Moo K. Chung.

**Data curation:** Shih-Gu Huang, Ian C. Carroll.

**Formal analysis:** Moo K. Chung, Shih-Gu Huang.

**Funding acquisition:** Moo K. Chung, H. Hill Goldsmith.

**Investigation:** Moo K. Chung, Shih-Gu Huang, Ian C. Carroll.

**Methodology:** Moo K. Chung, Shih-Gu Huang.

**Project administration:** Moo K. Chung, H. Hill Goldsmith.

**Resources:** Ian C. Carroll, H. Hill Goldsmith.

**Software:** Moo K. Chung, Shih-Gu Huang, Ian C. Carroll.

**Supervision:** Moo K. Chung, H. Hill Goldsmith.

**Validation:** Moo K. Chung.

**Visualization:** Moo K. Chung.

**Writing – original draft:** Moo K. Chung, Shih-Gu Huang.

**Writing – review & editing:** Moo K. Chung, Shih-Gu Huang, Vince D. Calhoun, H. Hill Goldsmith.

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
