## [Decision Letter · Decision Letter 0]

27 Nov 2023

Dear Dr. Chung,

Thank you very much for submitting your manuscript "Persistent Homological State-Space Estimation of Functional Human Brain Networks at Rest" for consideration at PLOS Computational Biology. As with all papers reviewed by the journal, your manuscript was reviewed by members of the editorial board and by several independent reviewers. The reviewers appreciated the attention to an important topic. Based on the reviews, we are likely to accept this manuscript for publication, providing that you modify the manuscript according to the review recommendations.

Both reviewers are positive to the manuscript. Please carefully revise the manuscript accordingly.

Sincerely,

Yalin Wang

Guest Editor

PLOS Computational Biology

Thomas Serre

Section Editor

PLOS Computational Biology

Both reviewers are positive to the manuscript. Please carefully revise the manuscript accordingly.

Reviewer's Responses to Questions

**Comments to the Authors:**

Reviewer #1: This paper presented a new data driven topological data analysis approach for estimating state spaces in dynamically changing human functional brain networks of human. It penalizes the topological distance between networks and clusters dynamically changing brain networks into topologically distinct states. The authors reported that its method outperformed the widely used k-means clustering often used in estimating the state space in brain network, and it was applied to more accurately determine the state spaces of dynamically changing functional brain networks. In general, this is a nice contribution to functional human brain networks. The methods are effective, the paper is well organized, and the results are promising.

There are a couple of minor suggestions.

1) There are too few references in the last 3 years, please update more recent literatures in the introduction and discussion sections.

2) Some important formulas should be numbered, such as in Definition and Theorem.

Reviewer #2: In this manuscript, the authors propose a new method based on the topological data analysis (TDA) approach to study the changes in resting-state functional MRI (rs-fMRI)-derived dynamic brain networks. Several advances are introduced including the use of the Wasserstein distance between the networks and testing of heritability in the twin study. Overall, the manuscript presents an interesting and novel approach to study the dynamics of rs-fMRI. A lot of presented work is an extension of the work of the first author and his collaborators who have been developing the TDA methods for more than a decade. However, there are several points that need further clarification. I have a number of general comments summarized below.

General comments:

1. Overall premise of the methodology is justified by the use of the Wasserstein distance to measure the distance between graphs. On page 11, the definition of the distance combines the 0D and 1D topological features via the sum of the squared distances. Can you please justify this definition?

2. Sections 5.1 and 5.2 summarize the simulation results for “Testing for false positives” (5.1) and “Testing for false negatives” (5.2). Is my understanding incorrect that the proposed method shows worse performance than the k-means and hierarchical clustering?

3. Sections 6.2 and 6.3 seem to be misplaced as they describe the methods used and not the data application.

4. Overall, the manuscript is written in a very technical way introducing several advanced mathematical concepts. An algorithmic summary of the steps necessary to apply the proposed method to the rs-fMRI data would add to a greater accessibility for a wider audience.

5. Some of the more technical material reviewing the existing concepts and methods can be moved to an appendix.

**Have the authors made all data and (if applicable) computational code underlying the findings in their manuscript fully available?**

Reviewer #1: Yes

Reviewer #2: **No: **Code has been made available, but the data have not been made public.

PLOS authors have the option to publish the peer review history of their article (what does this mean?). If published, this will include your full peer review and any attached files.

Reviewer #1: No

Reviewer #2: No

Figure Files:

Data Requirements:

Reproducibility:

References:

---

## [Decision Letter · Decision Letter 1]

29 Jan 2024

Dear Dr. Chung,

We are pleased to inform you that your manuscript 'Persistent Homological State-Space Estimation of Functional Human Brain Networks at Rest' has been provisionally accepted for publication in PLOS Computational Biology.

Best regards,

Yalin Wang

Guest Editor

PLOS Computational Biology

Thomas Serre

Section Editor

PLOS Computational Biology

The reviewers were satisfactory with the revision. It is a manuscript with good quality. It will have good impact to our research field.

Reviewer's Responses to Questions

**Comments to the Authors:**

Reviewer #1: The authors have solved all the concerns and thus I agree to accept it.

Reviewer #2: All comments have been addressed satisfactorily.

**Have the authors made all data and (if applicable) computational code underlying the findings in their manuscript fully available?**

Reviewer #1: Yes

Reviewer #2: **No: **The authors state: "Due to IRB protocol, data is not available. However, we made the code available at

https://github.com/laplcebeltrami/PH-STAT. For data access, please contact Moo K. Chung (first author) at mkchung@wisc.edu"

PLOS authors have the option to publish the peer review history of their article (what does this mean?). If published, this will include your full peer review and any attached files.

Reviewer #1: **Yes: **Liqun Kuang

Reviewer #2: No

---

## [Editor Report · Acceptance letter]

6 May 2024

PCOMPBIOL-D-23-01210R1 

Persistent Homological State-Space Estimation of Functional Human Brain Networks at Rest

Dear Dr Chung,

I am pleased to inform you that your manuscript has been formally accepted for publication in PLOS Computational Biology. Your manuscript is now with our production department and you will be notified of the publication date in due course.

With kind regards,

Anita Estes
